# Antimicrobial activity of iron-depriving pyoverdines against human opportunistic pathogens

Vera Vollenweider[1]*, Karoline Rehm[2], Clara Chepkirui[3], Manuela Pérez-Berlanga[1], Magdalini Polymenidou[1], Jörn Piel[3], Laurent Bigler[2], Rolf Kümmerli[1]*

[1]Department of Quantitative Biomedicine, University of Zurich, Zurich, Switzerland; [2]Department of Chemistry, University of Zurich, Zurich, Switzerland; [3]Institute of Microbiology, Eidgenössische Technische Hochschule (ETH) Zurich, Zurich, Switzerland

## eLife Assessment

This **important** study highlights the use of siderophores as antibacterials, and the authors also discuss the consequences and efficacy of 'siderophore therapy' in more complex communities/environments. The evidence supporting the overall hypotheses ranges is largely **convincing**. The work will be of broad interest to people working in the fields of evolutionary ecology, microbiology and medical sciences.

*For correspondence:
vera.vollenweider@uzh.ch (VV);
rolf.kuemmerli@uzh.ch (RK)

Competing interest: The authors declare that no competing interests exist.

**Abstract** The global rise of antibiotic resistance calls for new drugs against bacterial pathogens. A common approach is to search for natural compounds deployed by microbes to inhibit competitors. Here, we show that the iron-chelating pyoverdines, siderophores produced by environmental *Pseudomonas* spp., have strong antibacterial properties by inducing iron starvation and growth arrest in pathogens. A screen of 320 natural *Pseudomonas* isolates used against 12 human pathogens uncovered several pyoverdines with particularly high antibacterial properties and distinct chemical characteristics. The most potent pyoverdine effectively reduced growth of the pathogens *Acinetobacter baumannii*, *Klebsiella pneumoniae,* and *Staphylococcus aureus* in a concentration- and iron-dependent manner. Pyoverdine increased survival of infected *Galleria mellonella* host larvae and showed low toxicity for the host, mammalian cell lines, and erythrocytes. Furthermore, experimental evolution of pathogens combined with whole-genome sequencing revealed limited resistance evolution compared to an antibiotic. Thus, pyoverdines from environmental strains have the potential to become a new class of sustainable antibacterials against specific human pathogens.

## Introduction

There are approximately 4.95 million fatalities associated with antibiotic-resistant bacteria worldwide each year (**Murray et al., 2022**). This high mortality rate contrasts with the low development rate of new antibacterial agents for clinical applications (**WHO, 2022**; **Beyer and Paulin, 2020**). Thus, the development of alternative therapies with higher sustainability to counteract the rapidly dwindling treatment options due to resistance evolution is imperative (**Bell and MacLean, 2018**; **Monserrat-Martinez et al., 2019**; **Rezzoagli et al., 2020b**). Numerous alternative approaches have been proposed, including phage therapy, antimicrobial peptides and nanobodies, antivirulence treatments to disarm pathogens, treatments that capitalize on ecological competition between susceptible and resistant bacteria, and the search for new antibacterial compounds produced by environmental

**eLife digest** Despite the wide range of available antibiotics, a majority work through a similar mechanism, which enables some bacteria to become resistant to medical treatment. This means that the effectiveness of these molecules goes down and many curable infections may no longer be treatable.

Collectively, antibiotic resistance poses a similar threat as other major diseases, such as malaria. One promising approach for discovering new antibiotics is to explore natural microbial communities for their ability to produce secondary metabolites with antimicrobial properties. Such metabolites are typically secreted by bacteria to compete with other community members for essential resources including nutrients.

A well-known group of secreted metabolites are siderophores, which tightly sequester iron, a critical nutrient for bacterial growth. Each bacterial species produces its own set of specific siderophores, thus leading to a severe competition for iron.

Vollenweider et al. investigated whether a group of siderophores secreted by *Pseudomonas* bacteria, called pyoverdines, are an effective antimicrobial agent against harmful human pathogens. Pyoverdines have a high affinity to iron and prevent competing bacteria from accessing the critical nutrient. This can inhibit their growth by starving them of iron.

Vollenweider et al. treated resistant pathogens like *Acinetobacter*, *Klebsiella* and *Staphylococcus* grown in the laboratory, and found that pyoverdines significantly reduce their growth without the bacteria acquiring resistance. To test whether this treatment would work in a living infected animal, the group administered pyoverdines to moth larvae infected with the same pathogens and observed increased survival rates in the host.

As iron is also required for human metabolism and found in the haemoglobin of red blood cells, Vollenweider, et al. confirmed pyoverdines do not retrieve iron from haemoglobin. Finally, in laboratory settings, pyoverdines did not negatively affect the growth of a human and a mouse cell line at low concentrations strong enough to inhibit the growth of pathogens.

This approach is a promising example of adopting natural mechanisms that can have antimicrobial properties, and siderophores, in particular pyoverdines, may become a useful tool to treat otherwise incurable infections. Further research is needed in living mammalian models to confirm efficacy and safety of this novel antimicrobial treatment.

microbes (*Bhushan et al., 2017*; *Ghosh et al., 2019*; *Mei et al., 2022*; *Rezzoagli et al., 2020a*; *Saeki et al., 2020*; *Wale et al., 2017*).

Mining natural microbial communities has become a promising endeavor in the hunt for novel antimicrobials because most microbes deploy bioactive compounds to contend with their competitors in the diverse assemblies they live in. Competition often involves the secretion of secondary metabolites with specific antimicrobial properties against other microbes (*Hug et al., 2020*). The isolation, characterization, and synthesis of such secondary metabolites can result in novel prospective antibiotics for clinical applications (*Imai et al., 2019*; *Ling et al., 2015*). Secondary metabolites that act as toxins to kill competitors are often considered promising candidates. However, microbes compete through a variety of mechanisms other than toxins. For example, there is increasing evidence that competition for iron is a main determinant of species interactions in soil and freshwater communities (*Butaitè et al., 2017*; *Deveau et al., 2016*). Competition for iron involves the secretion of siderophores, a class of secondary metabolites that scavenges environmental iron with high affinity (*Hider and Kong, 2010*). Because siderophores are often species specific, they have two opposing effects: they make iron available for related community members possessing the matching receptor but withhold iron from competitors with non-matching receptors. Thus, siderophores can be competitive agents in interspecies interactions (*Figueiredo et al., 2022*; *Gu et al., 2020*; *Lee et al., 2012*; *Niehus et al., 2017*).

Here, we apply the concept of siderophore-mediated iron competition observed in natural communities to human opportunistic pathogens. Specifically, we investigate whether siderophores from non-pathogenic environmental bacteria can induce iron starvation and growth arrest in human pathogens. We focus on pyoverdines, a class of siderophores with high iron affinity that are produced and secreted by fluorescent *Pseudomonas* spp. under iron-limited conditions (*Visca et al., 2007*).

Upon complexation with ferric iron, the iron-loaded pyoverdine is imported into the cell by receptors with high specificity and subsequently reduced to the bio-available ferrous iron (*Bonneau et al., 2020*; *Cornelis et al., 2023*). Pyoverdines show an extraordinary structural diversity with more than 70 described variants that differ in their peptide backbone (*Meyer et al., 2008*; *Rehm et al., 2022*) and have an extremely high affinity for iron ($K_a = 10^{32}$ M$^{-1}$, pyoverdine produced by *Pseudomonas fluorescens* biotype B) (*Meyer and Abdallah, 1978*). For these reasons, we propose that pyoverdines from non-pathogenic *Pseudomonas* spp. could be potent agents to reduce the growth of opportunistic human pathogens by intensifying iron competition.

To test our hypothesis, we screened pyoverdines from a library of 320 environmental *Pseudomonas* strains, isolated from soil and freshwater habitats, for their activity against 12 strains of human opportunistic bacteria. For the most promising pyoverdine candidates, we elucidated the chemical structure to assess diversity and understand the chemical properties important for iron competition. Subsequently, we assessed the efficacy of the three top pyoverdine candidates against four human opportunistic pathogens (*Acinetobacter baumannii*, *Klebsiella pneumoniae*, *Pseudomonas aeruginosa*, and *Staphylococcus aureus*) in vitro and/or in vivo via infections of the Greater wax moth larvae (*Galleria mellonella*). We further assessed their toxicity in the host, toward two mammalian cell lines and red blood cells. Finally, we used experimental evolution combined with whole-genome sequencing to study the potential of pathogens evolving resistance against pyoverdine treatment.

## Results

### Pyoverdines can inhibit the growth of human opportunistic pathogens

To assess the antibacterial properties of pyoverdines, we performed a growth inhibition screen using a well-characterized collection of 320 natural *Pseudomonas* isolates (*Butaitė et al., 2017*; *Butaitė et al., 2018*; *Butaitė et al., 2021*; *Kramer et al., 2020*). The isolates stem from freshwater and soil communities and belong to the group of fluorescent pseudomonads (e.g., *P. fluorescens*, *P. putida*, *P. syringae*, *P. chlororaphis*) that are non-pathogenic to humans and can produce and secrete pyoverdines, a group of high iron-affinity siderophores (*Meyer, 2000*). Important to note is that several isolates are likely to produce secondary siderophores in addition to pyoverdine (*Cornelis, 2010*). Given that secondary

**Table 1.** Species and strains of human opportunistic pathogens used for the pyoverdine growth inhibition assay.

| Species and strain name | Class | Gram staining | Frequency of inhibiting supernatants |
|---|---|---|---|
| *Acinetobacter johnsonii* | Gammaproteobacteria | Negative | 75.0 |
| *Acinetobacter junii* | Gammaproteobacteria | Negative | 73.1 |
| *Burkholderia cenocepacia* H111 | Betaproteobacteria | Negative | 52.2 |
| *Burkholderia cenocepacia* K56-2 | Betaproteobacteria | Negative | 54.7 |
| *Cronobacter sakazakii* | Gammaproteobacteria | Negative | 48.8 |
| *Escherichia coli* K12 | Gammaproteobacteria | Negative | 31.6 |
| *Klebsiella michiganensis* | Gammaproteobacteria | Negative | 30.9 |
| *Pseudomonas aeruginosa* PA14 | Gammaproteobacteria | Negative | 9.7 |
| *Pseudomonas aeruginosa* PAO1 | Gammaproteobacteria | Negative | 20.9 |
| *Shigella* sp. | Gammaproteobacteria | Negative | 57.8 |
| *Staphylococcus aureus* Cowan | Bacilli | Positive | 47.8 |
| *Staphylococcus aureus* JE2 | Bacilli | Positive | 41.6 |

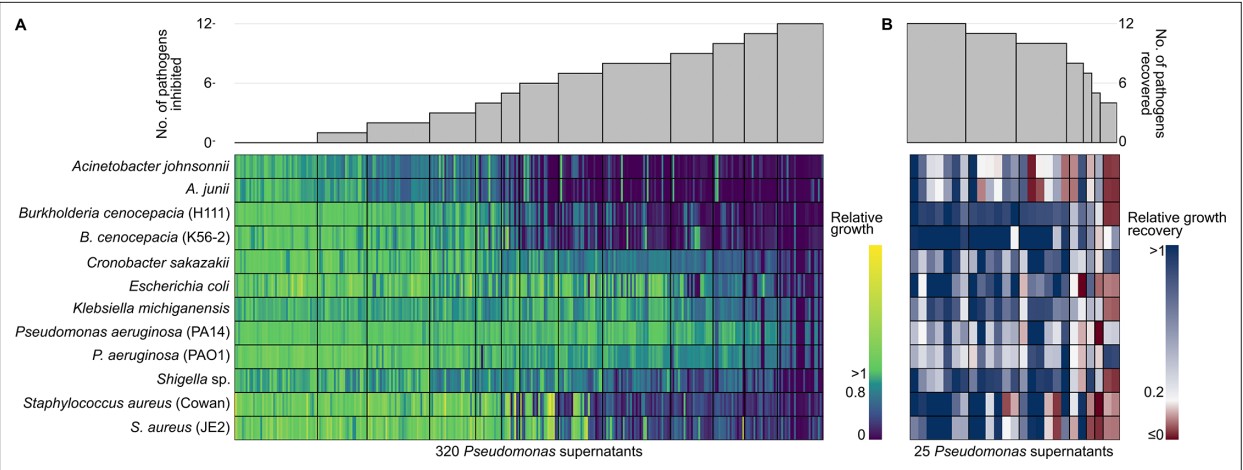

**Figure 1.** Effect of supernatants from environmental *Pseudomonas* isolates on the growth of 12 human opportunistic pathogens. (**A**) Screen to assess the extent to which pyoverdine-containing supernatants from 320 natural *Pseudomonas* isolates inhibit the growth of 12 human opportunistic pathogens. The heatmap depicts relative pathogen growth in the supernatant treatment (70% casamino acid medium [CAA] + 30% spent supernatant) relative to the control (70% CAA + 30% sodium chloride solution), with values ranging from stimulation (yellow) to inhibition (blue) based on four independent replicates. Gray bars above the heatmap show the number of pathogens that were at least 20% inhibited in their growth by a given supernatant. The screen returned 25 supernatant candidates that inhibited the growth of all pathogens. (**B**) Control screen with the 25 top supernatant candidates to check whether pyoverdine causes the observed growth inhibition. The heatmap depicts the level of growth recovery in the 12 pathogens when iron was added to the supernatant, with values ranging from no recovery (red) to full recovery (blue). Growth recovery in iron-rich medium indicates that pyoverdines are involved in growth inhibition in iron-limited medium. Gray bars above the heatmap show the number of pathogens that experienced a relative growth recovery of at least 0.2. The screen returned seven supernatant top candidates for which growth recovery occurred for all pathogens under iron-rich conditions.

The online version of this article includes the following figure supplement(s) for figure 1:

**Figure supplement 1.** Growth of pathogens in supernatant treatments and the number of pathogens inhibited by a supernatant.

**Figure supplement 2.** Cladogram of environmental *Pseudomonas* isolates from soil and pond habitats based on partial *rpoD* sequences.

siderophores generally have lower iron affinity than pyoverdine, we anticipate treatment effects to be primarily driven by pyoverdines.

In a first screen, we evaluated how the sterile pyoverdine-containing supernatants of the 320 isolates affect the growth of 12 human opportunistic pathogens (*Table 1*). Specifically, we grew the pathogens in 70% fresh casamino acid medium (CAA) supplemented with 30% spent supernatant. We found high variation in the supernatant effect, ranging from complete pathogen growth inhibition to growth promotion compared to the control treatment (70% CAA supplemented with 30% sodium chloride 0.8% solution) (*Figure 1A*, *Figure 1—figure supplement 1A*). The mean supernatant effect across all pathogens showed a bimodal distribution with two peaks at 0.47–0.58 (approximately 50% growth inhibition) and 1.00–1.10 (no growth inhibition) (*Figure 1—figure supplement 1B*). Our observations are consistent with previous studies, in which supernatants containing siderophores can have both growth inhibitory and stimulatory effects (*Gu et al., 2020*; *Butaitė et al., 2018*). For each supernatant, we quantified the number of pathogens that experienced a growth reduction of at least 20% compared to the control and found that 25 supernatants inhibited all 12 pathogens (*Figure 1A*, *Figure 1—figure supplement 1C*).

To confirm that the growth-inhibiting properties of the 25 most potent supernatants are indeed caused by pyoverdines and not by other agents in the supernatant, we repeated the above assay, but this time supplemented the supernatant with 40 µM FeCl$_3$. Under such iron-replete conditions, pyoverdines (and secondary siderophores) should not be able to substantially sequester environmental iron, and pathogen growth should therefore be restored. We identified seven top candidate supernatants for which a relative growth recovery of at least 0.2 occurred for all 12 pathogens (*Figure 1B*). Among the remaining 18 supernatants, the overall rate of growth recovery was also high, indicating that pyoverdines are responsible for pathogen growth inhibition under iron-limited conditions in most cases.

Finally, we assessed the relationship between growth inhibitory effects of supernatants and the phylogenetic affiliation of the producing strains (*Figure 1—figure supplement 2*). We found that growth inhibitory effects were not restricted to a few closely related strains but were scattered across the phylogenetic tree. Nonetheless, there was a certain level of clustering with two *Pseudomonas* spp. clades showing an accumulation of strains with growth-inhibitory effects on pathogens. This analysis suggests that closely as well as distantly related strains can produce inhibitory pyoverdines.

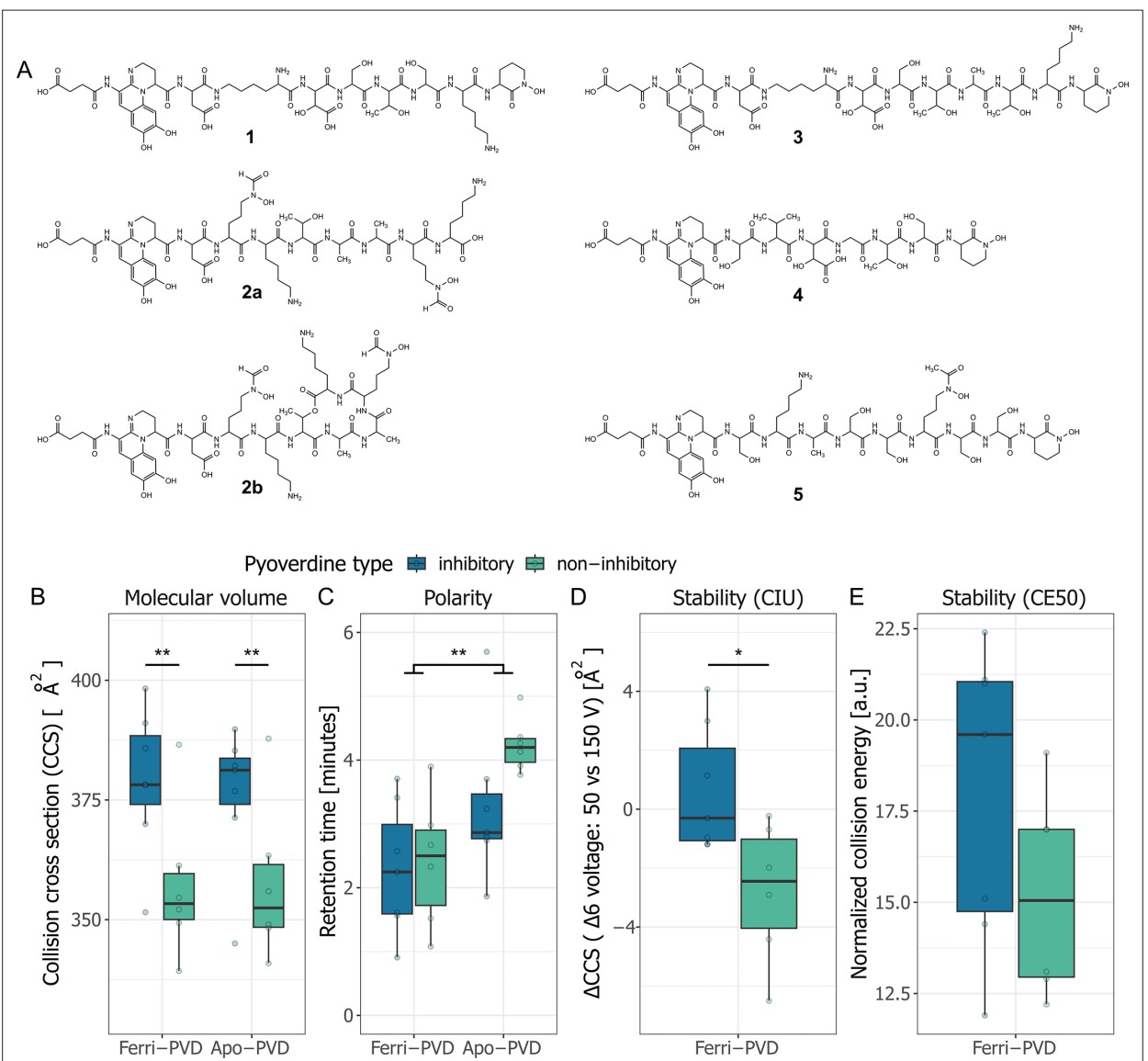

**Figure 2.** Chemical structure of growth-inhibitory pyoverdines and their properties compared to non-inhibitory pyoverdines. (**A**) The chemical structures of the seven top candidate pyoverdines were elucidated using ultra-high-performance liquid chromatography high-resolution tandem mass spectrometry (UHPLC-HR-MS/MS) and revealed five unique pyoverdine structures (**1–5**) differing in their peptide backbone (see *Rehm et al., 2022* for an in-depth chemical analysis). Pyoverdine **1** is a novel structure (from isolate 3A06). Pyoverdine **2** can occur in either a linear **2a** or a cyclic **2b** form (from isolate 3G07). Pyoverdine **3** was found in three different isolates, originating from the same soil sample (from isolates s3b09, s3b10, and s3b12). Pyoverdine **4** and **5** are from isolate s3c13 and s3e20, respectively. (**B**) The collision-cross sections (CCS) values of inhibitory pyoverdines are higher than for non-inhibitory pyoverdines. (**C**) Molecule polarity, measured as the chromatographic retention time, is higher for ferri-pyoverdines (iron loaded) than for apo-pyoverdines (iron free). (**D**) Iron complex stability, assessed by the collision-induced unfolding (CIU) of ferri-pyoverdines, was significantly higher for inhibitory than non-inhibitory pyoverdines. (**E**) The normalized collision energy (CE50) necessary to fragment 50% of ferri-pyoverdines was not different between inhibitory and non-inhibitory pyoverdines. Box plots show the median and the first and third quartiles across the seven inhibitory and six non-inhibitory pyoverdines. Whiskers represent the 1.5× interquartile range. Significance levels are based on ANOVAS (*$p < 0.05$ and **$p < 0.01$).

## Growth-inhibiting pyoverdines show distinct chemical features

In previous studies, we developed a new method for pyoverdine structure elucidation from low-volume crude extracts using ultra-high-performance liquid chromatography high-resolution tandem mass spectrometry (UHPLC-HR-MS/MS) (*Rehm et al., 2022*; *Rehm et al., 2023*). These studies included the pyoverdines extracted from the seven top candidate supernatants and the analysis therein yielded five structurally different pyoverdines (*Figure 2A*). We found one novel pyoverdine type (**1**, from isolate 3A06) and four pyoverdine types already known from the literature (**2–5**). The pyoverdine from isolate 3G07 can occur in either a linear or cyclic conformation (**2a** and **2b**). Three isolates (s3b09, s3b10, s3b12), all originating from the same soil sample, had an identical pyoverdine type (**3**).

Here, we investigated whether the chemical properties of growth-inhibitory pyoverdines differ from pyoverdines that do not inhibit pathogen growth. To this end, we further elucidated the chemical structure of a set of six pyoverdines extracted from supernatants that inhibited none of the 12 pathogens, and one additional pyoverdine inhibiting the growth of 10 pathogens (*Rehm et al., 2022*). We found that inhibitory pyoverdines had significantly larger collision-cross sections (CCS in Å$^2$) than non-inhibitory ones ($F_{1,22}$ = 10.63, p=.0036, measured by ion mobility spectrometry [IMS]) (*Figure 2B*). As CCS values correlate with the volume of a molecule, our results suggest that inhibitory pyoverdines are larger and have more complex molecule structures than non-inhibitory pyoverdines. In contrast, there was no significant difference between the two classes of pyoverdines regarding molecule polarity ($F_{1,22}$ = 2.00, p=0.1711, assessed by the chromatographic retention time [RT] on a reversed-phase column) (*Figure 2C*), but molecule polarity was significantly higher for ferri- than for apo-pyoverdines ($F_{1,22}$ = 12.78, p=0.0017). Finally, we assessed the stability of ferri-pyoverdines to derive a proxy for iron affinity. We first measured the propensity of ferri-pyoverdines to undergo collision-induced unfolding (CIU) by IMS measurements at different Δ6 voltages and found that inhibitory pyoverdines are significantly more stable than non-inhibitory pyoverdines ($F_{1,11}$ = 7.55, p=0.019) (*Figure 2D*). The same trend was observed for our second measure of stability (CE50, normalized collision energy (NCE) necessary to fragment 50% of ferri-pyoverdines), but in this case, the difference between the two pyoverdine classes was not significant ($F_{1,11}$ = 1.87, p=0.198) (*Figure 2E*). Taken together, the chemical analyses suggest that inhibitory pyoverdines are larger and have a higher affinity for iron than non-inhibitory pyoverdines.

## Pyoverdines inhibit human pathogens in a concentration-dependent manner

We purified three out of the five most potent pyoverdines and tested their efficacy against the four human pathogens *A. baumannii*, *K. pneumoniae*, *P. aeruginosa* PAO1, and *S. aureus* JE2. For these experiments, we decided to include the most common (s3b09), the novel (3A06), and the cyclical-linear (3G07) pyoverdines. We crude-purified all three pyoverdines using established protocols (*Butaitė et al., 2017*; *Meyer et al., 1997*) and further purified the most common pyoverdine (s3b09) using preparative and analytical HPLC. During HPLC purification, we observed that ferribactin (the final pyoverdine precursor) was also highly abundant in the supernatant (*Budzikiewicz et al., 2007*; *Nadal-Jimenez et al., 2014*), presumably released by cells during the centrifugation process. This precursor is typically not secreted but matures into pyoverdine in the periplasma (*Sugue et al., 2022*). Due to its compromised iron-binding capacity, we used it as a negative control for which no activity against pathogens is expected (*Nadal-Jimenez et al., 2014*). As positive treatment control, we used the antibiotic ciprofloxacin. We subjected the four pathogens to the three pyoverdine variants, ferribactin, and ciprofloxacin across a concentration gradient and measured their effects on pathogen growth (*Figure 3*, *Figure 3—figure supplement 1*).

For *A. baumannii* and *S. aureus,* we observed classical dose–response curves in all pyoverdine assays, characterized by a decrease in bacterial growth at low and intermediate concentrations, followed by a complete halt of bacterial growth at higher pyoverdine concentrations. There was a good qualitative match between the dose–response curves obtained with the HPLC vs. the crude-purified pyoverdines. Note that pyoverdine concentration is unknown for the crude extracts and is thus expressed as relative concentrations. In contrast, the pyoverdine concentration is absolute for the HPLC-purified version, allowing the determination of an IC50 value for pyoverdine s3b09, which is 5.037 µg/mL and 12.163 µg/mL for *A. baumannii* and *S. aureus*, respectively.

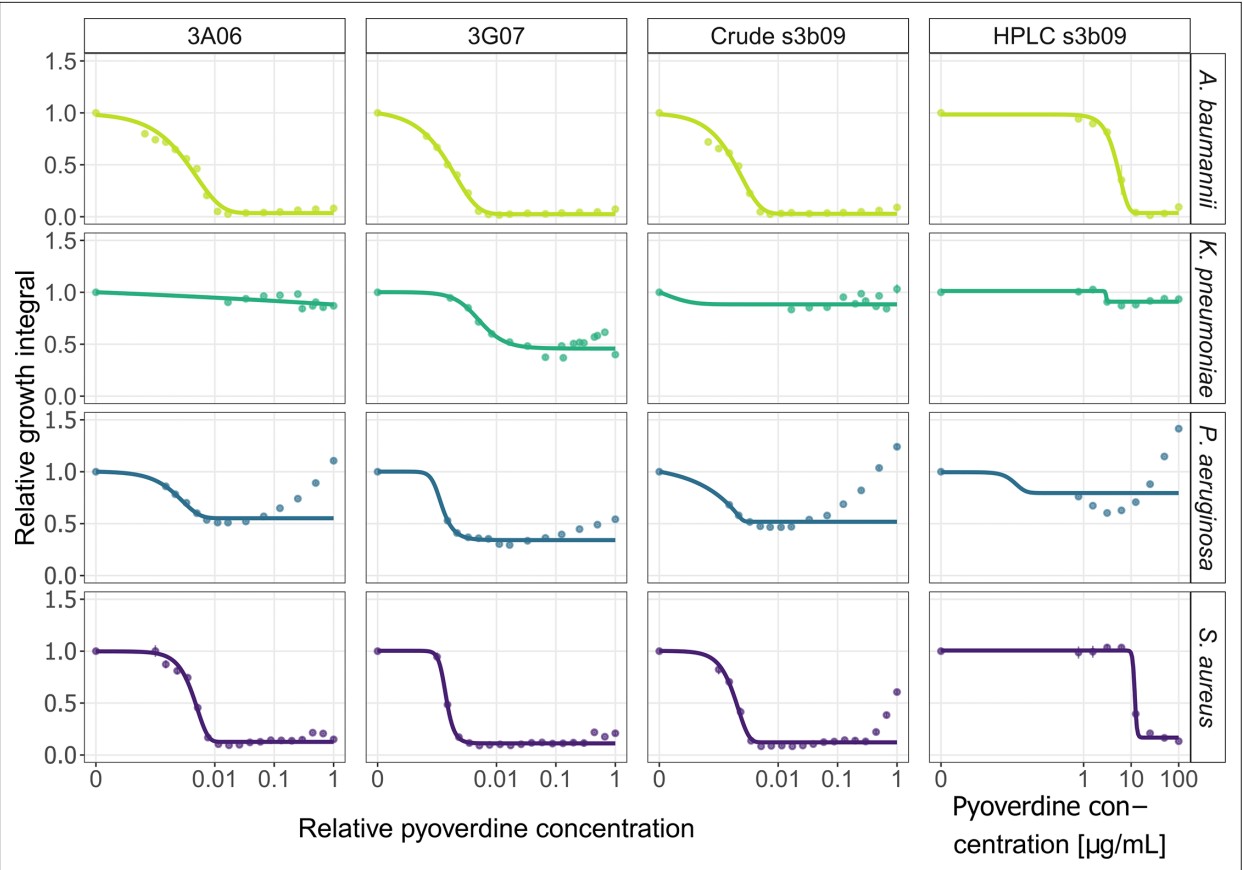

**Figure 3.** Pyoverdine dose–response curves for *A. baumannii*, *K. pneumoniae*, *P. aeruginosa*, and *S. aureus*. We exposed the four human opportunistic pathogens to three pyoverdines (3A06, 3G07, s3b09) that were among the most potent ones. We used crude-purified extracts of all three pyoverdines and a high-performance liquid chromatography (HPLC)-purified variant for pyoverdine s3b09. The absolute concentrations of the crude-purified extracts are unknown and therefore expressed relative to the weighed amount of 6 mg. The absolute concentration of the HPLC-purified variant is given in µg/mL. Growth values are scaled relative to the untreated control in casamino acid medium (CAA) medium. Dots and error bars show mean values and standard errors, respectively, across a minimum of three replicates per concentration. Dose–response curves were fitted using four- or five-parameter logistic regressions.

The online version of this article includes the following figure supplement(s) for figure 3:

**Figure supplement 1.** Ferribactin and ciprofloxacin dose-response curves for *A. baumannii*, *K. pneumoniae*, *P. aeruginosa*, and *S. aureus*.

**Figure supplement 2.** The effect of iron-saturated pyoverdines on the growth of *A. baumannii*.

**Figure supplement 3.** High-resolution mass spectrum (HRMS) of purified pyoverdine s3b09.

**Figure supplement 4.** High-resolution mass spectrum (HRMS) of purified ferribactin (precursor molecule of pyoverdine s3b09).

For *K. pneumoniae*, the pyoverdines were less effective in reducing pathogen growth. The dose–response curves for the pyoverdines s3b09 (HPLC and crude) and 3A06 showed similar trajectories and the model fits showed a growth reduction between 9–12%. The strongest inhibitory effects were found for pyoverdine 3G07 with an estimated 54% growth reduction. One reason why pyoverdines might be less effective against *K. pneumoniae* is that this species produces enterobactin, a siderophore that has higher iron affinity than pyoverdine (*Harris et al., 1979*).

For *P. aeruginosa*, we observed that all pyoverdines were growth inhibitory at intermediate concentrations. The initial inhibition converted into growth promotions at high pyoverdine concentrations in three out of four cases. The only exception was pyoverdine 3G07, which yielded consistent growth inhibitions of about 66%. One reason for these non-standard dose–response curves could be that *P. aeruginosa* possesses two receptors (FpvA and FpvB) for pyoverdine uptake (*Ghysels et al., 2004*). It might thus capitalize on the supplemented pyoverdines, especially at high concentrations via the upregulation of the heterologous promiscuous FpvB receptor (*González et al., 2021*; *Sexton et al.,*

*2017*). Our findings for *P. aeruginosa* are not unexpected and show that pyoverdine treatment should not be applied against this pathogen.

Taken together, we found that all pyoverdines are highly potent against *A. baumannii* and *S. aureus*, and one pyoverdine (3G07) is potent against *K. pneumoniae*. Especially in the case of *A. baumannii* and *S. aureus*, the pyoverdine dose–response curves follow the standard trajectory observed for antibiotics (*Figure 3—figure supplement 1* for ciprofloxacin). In support of our hypothesis that the inhibitory effects of pyoverdines operate via withholding iron from pathogens, we found no growth inhibition when the pathogens were exposed to purified ferribactin (*Figure 3—figure supplement 1*), the non-iron chelating precursor of pyoverdine.

To further validate that the antibacterial effect of pyoverdine is due to iron sequestration, we saturated pyoverdine s3b09 with varying concentrations of iron and repeated the dose–response experiment with *A. baumannii* (*Figure 3—figure supplement 2*). Here, we expect pyoverdine potency to decrease because iron is no longer a growth limiting factor. Indeed, we observed that the inhibition of *A. baumannii* was greatly reduced with intermediate levels of iron supplementation (40 μm) and completely stalled with high levels of iron supplementation (200 μM) (*Figure 3—figure supplement 2*).

## Pyoverdines have low toxicity for mammalian cell lines, erythrocytes, and hosts

We investigated the potential toxicity of crude-purified pyoverdines against mouse neuroblastoma-spinal cord (NSC-34) hybrid cells and human embryonic kidney 293 (HEK-293) cells. NSC-34 are motor neuron-like cells, which do not divide in their differentiated state, thus mimicking established tissue. For these cells, we observed no adverse effects on cell viability at low and intermediate pyoverdine concentrations (*Figure 4A*), representing the doses that strongly inhibited the pathogens (*Figure 3*). A moderate decrease in cell viability was only observed at the highest pyoverdine concentrations (*Figure 4A*). Conversely, HEK-293 cells divide rapidly and inform us on whether pyoverdines interfere with cell proliferation. We indeed noticed a more pronounced effect on cell viability already at intermediate pyoverdine concentrations (*Figure 4A*), which suggests that pyoverdine chelates the iron required for cell proliferation.

To assess the hemolytic activity of pyoverdines, we exposed sheep erythrocytes to a concentration gradient of the crude-purified pyoverdines. We found that hemolysis was extremely low (<1%) relative to the positive control Triton X-100 surfactant for pyoverdine concentrations up to 0.5 (*Figure 4B*) and remained low even for the highest concentration (hemolysis rates for pyoverdines 3A06, 3G07, and s3b09 were 2.01, 1.89, and 5.92%, respectively). This result shows that pyoverdines are unable to retrieve iron from hemoglobin.

Finally, we injected pyoverdine into the larvae of *G. mellonella* and followed host survival over time. We selected three pyoverdine concentrations (0.01, 0.05, and 0.1), which we later used for the infection experiments, and which were minimally toxic to cell lines (colored dots in *Figure 4A*). We found that pyoverdine injections did not significantly reduce host survival at low and intermediate concentrations (*Figure 4C* and *Supplementary file 1a*). In contrast, we observed a tendency toward higher larval mortality levels with the highest pyoverdine concentration administered, especially for 3G07, suggesting that pyoverdines can be mildly toxic for the host at high concentrations.

## Pyoverdine treatment increases the survival of infected hosts

We then investigated whether pyoverdines can be used to treat infections in *G. mellonella*. We used *A. baumannii*, *K. pneumoniae*, and *P. aeruginosa* PAO1 as pathogens. We did not include *S. aureus* for these experiments as it is literally avirulent for *G. mellonella* given the infection loads and time frames used (*Ménard et al., 2021*). We first determined the bacterial infection load needed to reduce larval survival by at least 50% (*Figure 5—figure supplement 1*) over 48 hr. We found this to be the case with $1.8 * 10^5$ CFU/larvae of *A. baumannii* and $8.9 * 10^5$ CFU/larvae of *K. pneumoniae*. For *P. aeruginosa*, all larvae died within 48 hr even with the lowest infection dose and we decided to use 56 CFU/larvae. We infected the larvae with one of the three pathogens and subsequently treated them with either pyoverdine (relative concentrations 0.01, 0.05, and 0.1) or a PBS control 4 hr post infection, followed by monitoring larval survival over time (*Figure 5*, *Figure 5—figure supplement 2*).

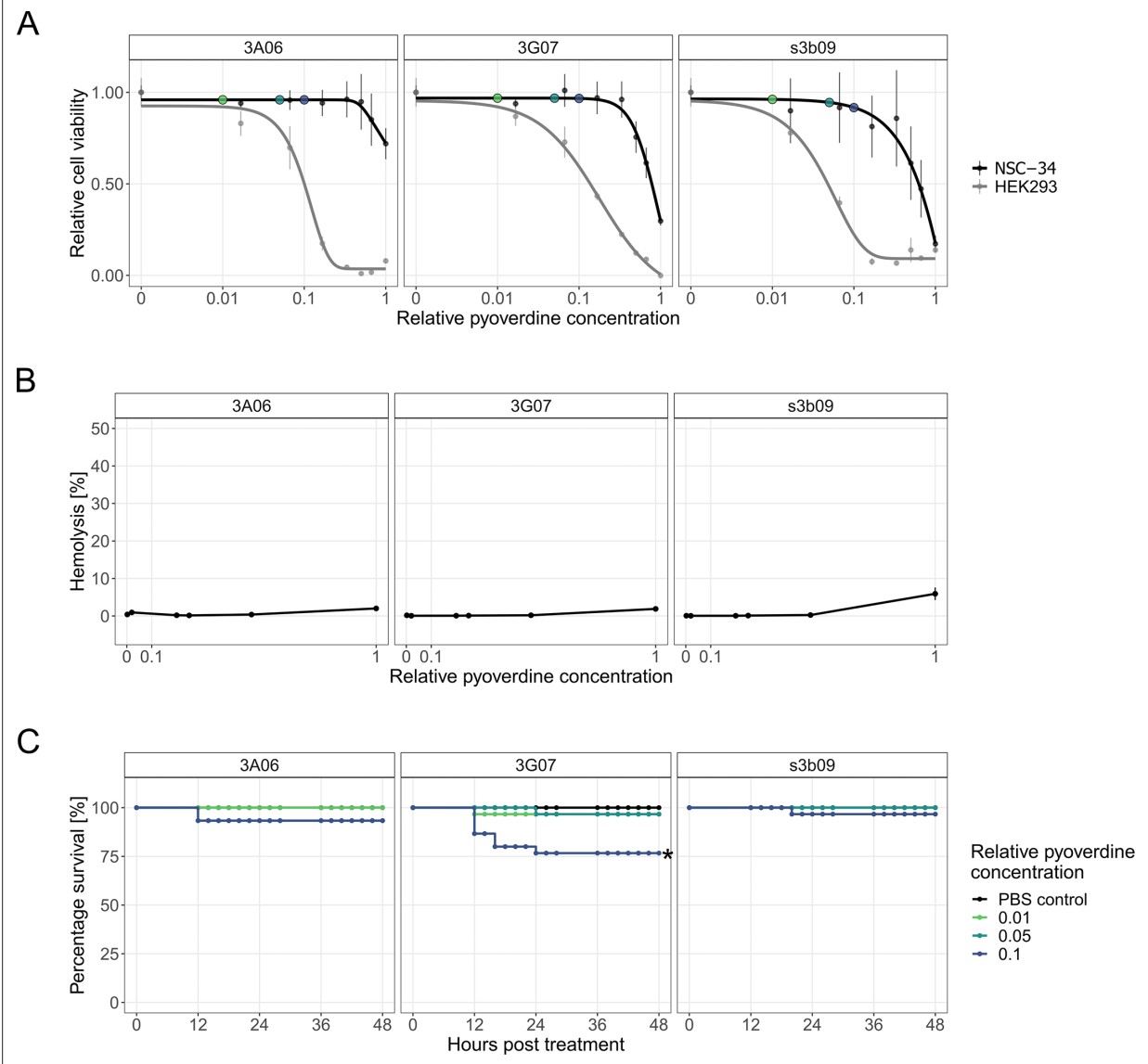

**Figure 4.** Toxicity assays for pyoverdines from environmental *Pseudomonas* spp. against human cell lines, sheep erythrocytes, and the host larvae of *G. mellonella*. (**A**) We exposed mouse neuroblastoma-spinal cord (NSC-34) and human embryonic kidney 293 (HEK-293) cells to three crude-purified pyoverdines (3A06, 3G07, s3b09) that were among the most potent ones to inhibit bacterial growth. An MTT assay was used to assess the metabolic activity of cells as an indicator of cell viability and proliferation. Cell viability data are scaled relative to the pyoverdine-free treatment, whereby dots and error bars show means and standard error across three replicates, respectively. The absolute concentrations of the crude-purified pyoverdines are unknown and concentrations are therefore expressed relative to the highest one used. Colored dots indicate relative pyoverdine dosages used for the in vivo experiments. Dose–response curves were fitted using five-parameter logistic regressions. (**B**) We evaluated the hemolytic activity of the pyoverdines by adding them to sheep erythrocytes along a concentration gradient (range 0.002–1). Triton X-100 and PBS served as positive and negative control, respectively. Hemolytic activity is scaled relative to the positive control, dots and error bars show means and standard error across six replicates from two independent experiments, respectively. (**C**) To assess the toxic effects of pyoverdine on the host, we injected pyoverdines (three relative concentrations, 0.01, 0.05, 0.1) into larvae 4 hr after a mock infection with PBS. The percentage of larval survival was tracked over 48 hr post-treatment. Data stem from three independent experiments with each 10 larvae per infection and treatment. Significance based on the log-rank test (adjusted for multiple comparisons using the Holm method).

We found that the two pyoverdines 3A06 and 3G07 were effective treatments against *A. baumannii* and *K. pneumoniae* and significantly decreased the risk of death compared to untreated infections (***Figure 5*** and ***Supplementary file 1b***). Specifically, *A. baumannii* treated with 3A06 at a concentration of 0.05 and 3G07 at a concentration of 0.01 reduced the risk of death (hazard) by 57.9% (z = −1.973, p=0.0485) and 59.4% (z = −2.057, p=0.0397), respectively. For s3b09, the same tendency was

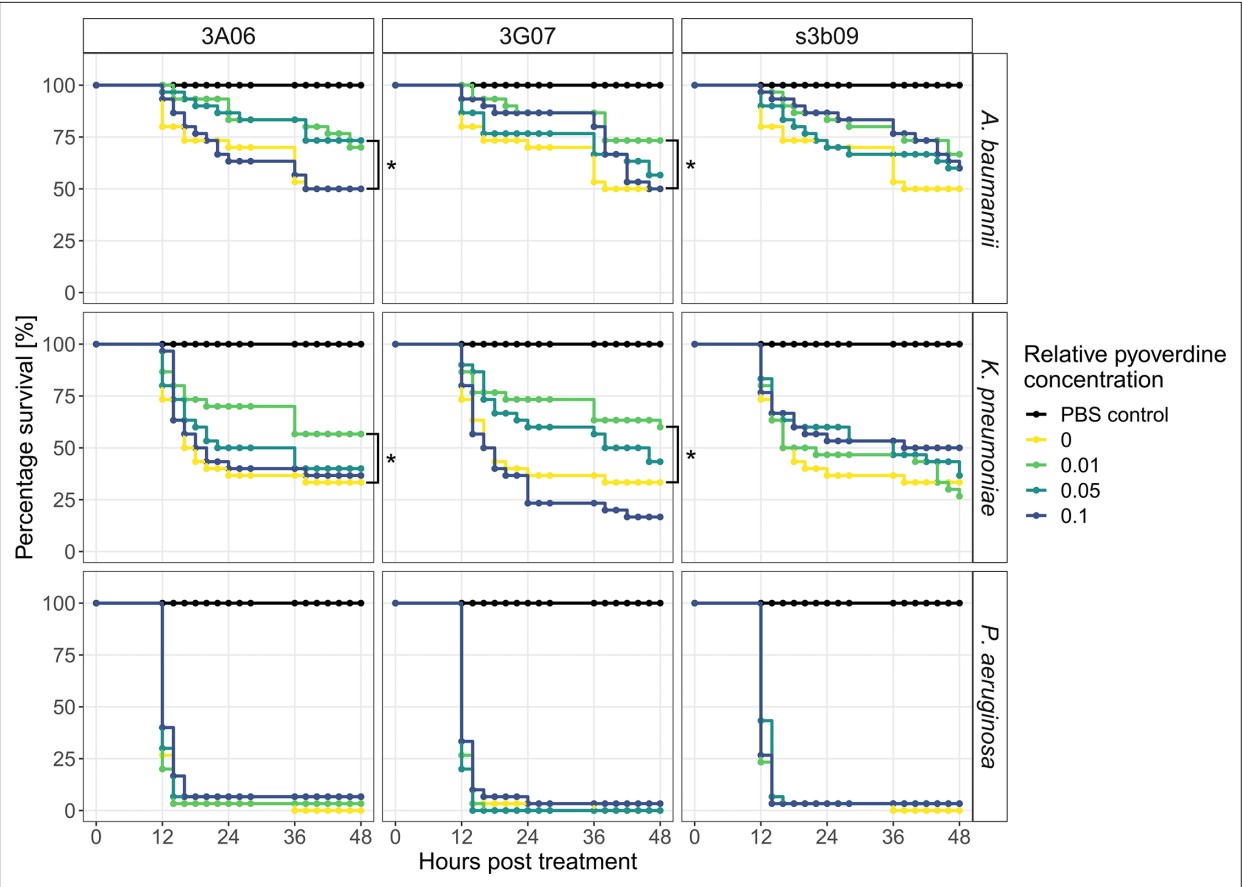

**Figure 5.** Pyoverdine treatments significantly increase survival of the host *G. mellonella* when infected with *A. baumannii* and *K. pneumoniae*. Larvae of the greater wax moth were first infected with either *A. baumannii*, *K. pneumoniae*, or *P. aeruginosa* and then treated with one of three pyoverdines (3A06, 3G07, or s3b09) at four relative pyoverdine concentrations (0, 0.01, 0.05, 0.1). All panels show the larvae survival over 48 hr post-treatment. Data stem from three independent experiments with each 10 larvae per infection and treatment. Asterisks indicate significant differences in larval survival between treated and untreated infections based on Cox proportional hazard regressions (p<0.05).

The online version of this article includes the following figure supplement(s) for figure 5:

**Figure supplement 1.** Survival curves of *G. mellonella* infected with *A. baumannii*, *K. pneumoniae,* or *P. aeruginosa* at different infection loads.

**Figure supplement 2.** Variation in the time-to-death of *G. mellonella* larvae in response to pyoverdine treatment.

observed (z = −1.496, p=0.135) albeit not significant. For *K. pneumoniae*, the pyoverdine treatments 3A06 and 3G07 (concentration of 0.01) reduced the risk of death by 51.5% (z = −2.027, p=0.0426) and 57.6% (z = −2.341, p=0.0192), respectively. These results suggest that low and intermediate doses of pyoverdine are more effective than the highest concentration. This notion was further supported when analyzing the time-to-death of larvae (*Figure 5—figure supplement 2*). We found hump-shaped relationships between pyoverdine concentrations and time-to-death in four out of six cases with two of them being significant (quadratic regression models for *K. pneumoniae* treated with pyoverdine 3G07: linear term: $F_{1,71}$ = 0.88, p=0.3506, quadratic term: $F_{1,71}$ = 6.11, p=0.0159 and s3b09: linear term: $F_{1,73}$ = 0.31, p=0.5806, quadratic term: $F_{1,73}$ = 6.16, p=0.0153). In contrast, infections with *P. aeruginosa* led to near 100% larval mortality regardless of pyoverdine treatment and dosage so that no significant treatment effect arose (*Figure 5*) and no effect on the time-to-death of larvae could be observed (*Figure 5—figure supplement 2*).

Altogether, our infection experiments show that pyoverdines are effective against the moderately virulent pathogens *A. baumannii* and *K. pneumoniae* by decreasing the risk of death by more than 50%, while they are ineffective against the fast-killing highly virulent *P. aeruginosa*. Moreover, intermediate pyoverdine concentration are most effective, reinforcing our observation that pyoverdine can exhibit some toxicity at higher concentrations (*Figure 4C*).

## Pathogens show low levels of resistance evolution against pyoverdine treatment

We assessed the ability of *A. baumannii*, *K. pneumoniae*, *P. aeruginosa*, and *S. aureus* to evolve resistance against the pyoverdine treatment by exposing all pathogens to both single pyoverdine treatments (3A06, 3G07, and s3b09) and combination treatments (double and triple combinations) over 16 days of experimental evolution. One exception was *K. pneumoniae*, which was only exposed to 3G07 as only this pyoverdine was inhibitory. We had six independent lineages (populations) per pathogen and treatment combination (see *Supplementary file 1b* for treatment concentrations). We further subjected all pathogens to the antibiotic ciprofloxacin as positive control in which resistance evolution is expected. Additionally, we let the pathogens grow in untreated growth medium to control for adaptation to the growth medium. Overall, we had 180 lineages that were transferred daily to fresh medium.

After experimental evolution, we observed that all four pathogens grew significantly better under ciprofloxacin treatment than the ancestor and beyond the level observed for growth medium adaptation (*Supplementary file 1c*; *Figure 6A*, *Figure 6—figure supplement 1*). This provides strong evidence for pervasive resistance evolution against this conventional antibiotic. In contrast, we found that none of the four pathogens treated with pyoverdines experienced a significant increase in growth relative to the observed level of growth medium adaptation (*Supplementary file 1c*, *Figure 6A*), suggesting low potentials of resistance evolution at the population level.

However, population screens can hide the presence of resistant clones within populations. We thus picked eight random clones from two populations for each of the pyoverdine single treatments and one combo treatment for each pathogen. With a total of 208 clones, we repeated the above growth assay and expressed clonal growth as fold-change relative to the growth of populations evolved in the growth medium alone (*Figure 6B*, *Figure 6—figure supplement 2*). For *A. baumannii* and *K. pneumoniae*, the average growth fold-change across clones was 1.02 ± 0.004 and 0.84 ± 0.02, respectively. This suggests that clones grew similar to the ancestors and that the variation among clones is low. This was different for *P. aeruginosa* and *S. aureus*, for which we observed high variation in growth fold-changes across clones, with many clones growing moderately better than the growth medium controls (*Figure 6B*, *Figure 6—figure supplement 2*). The growth increase across clones was 1.11 ± 0.02 and 1.10 ± 0.02, and significant in three and two population of *P. aeruginosa* and *S. aureus*, respectively. Thus, the clonal screen indicates both a certain heterogeneity among clones and an overall moderate adaptation to pyoverdine treatment for these two pathogens.

## Mutational and functional patterns in response to pyoverdine treatment are species specific

To obtain insights into the genetic basis of putative resistance evolution, we sequenced the whole genomes of 52 evolved clones (16 per pathogen, except 4 for *K. pneumoniae*), 16 medium-evolved control populations, and the 4 ancestors. We first mapped reads to the reference genomes and then to our ancestors. Among the clones evolved under pyoverdine treatment, we identified 97 synonymous, 96 nonsynonymous, 59 intergenic, and 9 nonsense single-nucleotide polymorphisms (SNPs). Additionally, there were 48 insertions, 59 deletions, and 45 substitutions (*Figure 6B*). We observed that growth peaked in clones with a single mutation and decreased in clones with multiple mutations (*Figure 6C*, linear fit: $F_{1,50} = 6.57$, p=0.0135), suggesting positive selection for a few specific advantageous mutations.

Next, we excluded all mutations that were present at least twice in the no-treatment controls, and all intergenic mutations. We identified 7 mutations in *A. baumannii*, 1 in *K. pneumoniae*, 10 in *P. aeruginosa*, and 20 in *S. aureus* that exclusively arose under pyoverdine treatment and could thus be associated with resistance evolution. We then classified the SNPs and small deletions (<600 bp) into functional categories (*Figure 6D* and *Supplementary file 1d*). We found that mutations in genes related to regulation and metabolism were consistently associated with increased growth in *S. aureus*, whereas mutations in genes related to biofilm and motility were associated with increased growth in *P. aeruginosa* (*Figure 6D*).

Furthermore, we identified six large deletions (≥600 bp; *Supplementary file 1e*). One *K. pneumoniae* clone lost an ~42 kb plasmid encoding various functional proteins, such as methyltransferases and toxin-antitoxin systems (*Singh et al., 2021*). Four *S. aureus* clones from two independent populations

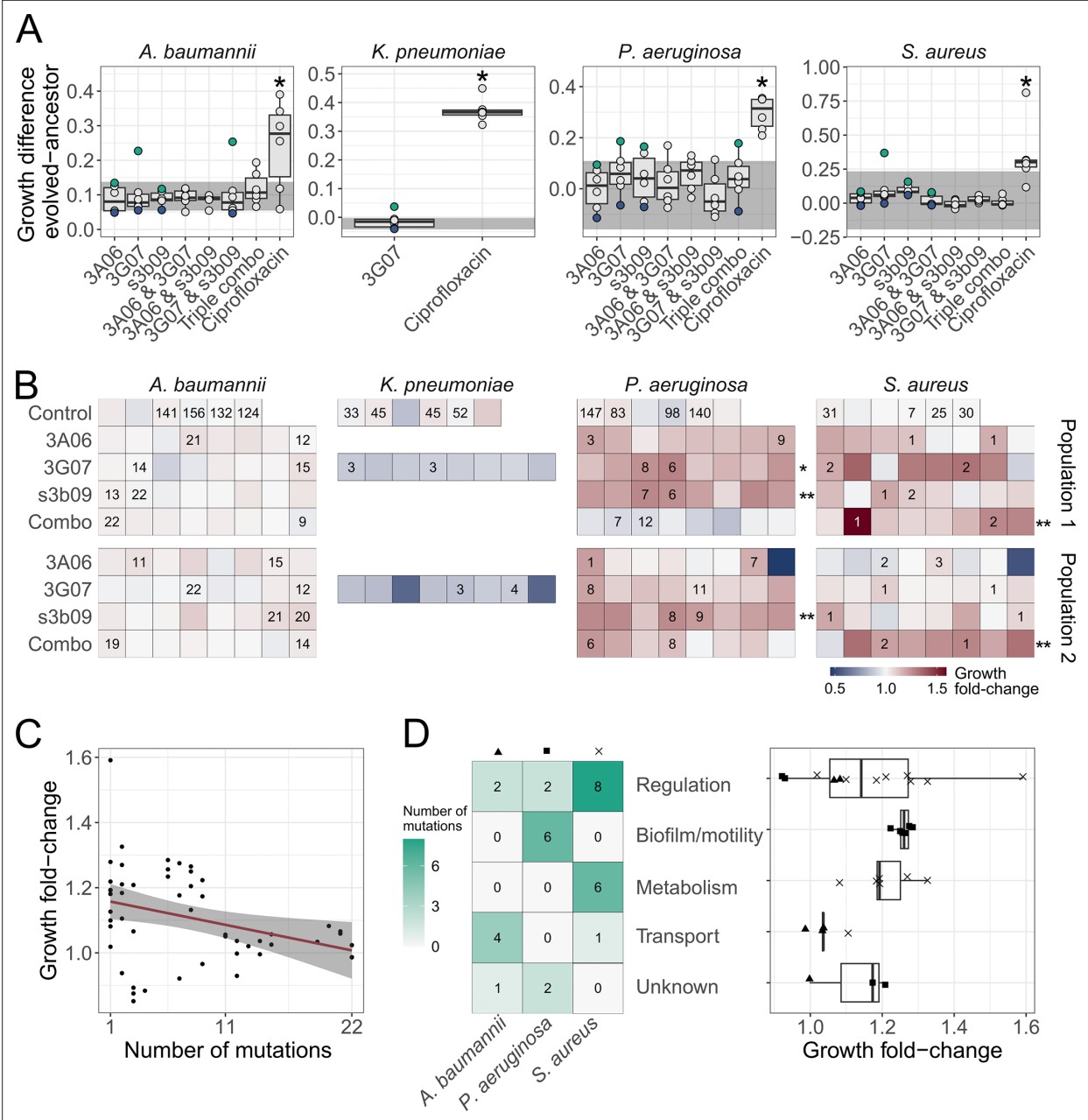

**Figure 6.** Phenotypic and genotypic analysis of experimentally evolved pathogens reveal weak levels of resistance evolution against pyoverdine treatment. (**A**) We exposed evolved and ancestral pathogen populations to the treatment in which they evolved in and quantified their growth (area under the curve). Growth values were scaled relative to the ancestor in untreated medium, and the panels show the scaled growth differences between evolved and ancestral populations. The shaded areas show the scaled growth difference between the ancestor and the populations evolved in untreated medium and is representative of medium adaptation. The blue and green dots represent the pathogen populations with the lowest (population 1) and highest (population 2) scaled growth difference, respectively, which were subsequently used to pick clones. The dots show mean values across two independent replicates and asterisks show significant growth increases relative to the medium-adapted control. Box plots show the median and the first and third quartiles across the six independently evolved populations. Whiskers represent the 1.5× interquartile range. Significance levels are based on two-sided t-tests, Welch's t-tests, or Wilcoxon tests (*p < 0.05) (**B**) We repeated the above growth assays with 208 individual clones evolved under the pyoverdine treatments (picked from population 1 and population 2) and 24 populations evolved in growth medium alone (control). Each square represents a clone or a control population, and the number indicates the number of mutations identified based on whole-genome sequencing. The heatmap shows the fold-change in growth relative to the evolved control populations. Asterisks depict significant fold-increases in growth compared to control populations (based on one- and two-way ANOVAs, where *p < 0.05 and **p < 0.01). (**C**) The relationship between the number of mutations and fold-change in growth across all sequenced clones (n = 52) The red line and shaded area are the regression line and 95% confidence interval, respectively. (**D**) Heatmap showing the number of mutations per pathogen and per functional gene categories across clones

*Figure 6 continued on next page*

*Figure 6 continued*

together with the respective fold-change in growth (n = 32). For this analysis, we excluded intergenic mutations and deletions larger than 600 bp. Triangle, square, and cross depict growth fold-change of *A. baumannii*, *P. aeruginosa,* and *S. aureus*, respectively.

The online version of this article includes the following figure supplement(s) for figure 6:

**Figure supplement 1.** Population growth kinetics of the pathogens *A. baumannii*, *K. pneumoniae*, *P. aeruginosa*, and *S. aureus* before and after experimental evolution.

**Figure supplement 2.** Growth kinetics of evolved clones of the pathogens *A. baumannii*, *K. pneumoniae*, *P. aeruginosa,* and *S. aureus* in comparison to the ancestor.

showed parallel loss of the ~14 kb pathogenicity island SaPI5, containing the virulence genes *sek* and *seq* (*Diep et al., 2006*; *Novick and Ram, 2017*). Additionally, one *S. aureus* clone exhibited an ~55 kb deletion containing the staphylococcal cassette chromosome mec type IV (SCCmecIV), which harbors the *mecA* gene responsible for β-lactam antibiotic resistance, and the arginine catabolic mobile element (ACME) (*Diep et al., 2006*).

Taken together, we found that the observed weak resistance evolution against pyoverdine in *P. aeruginosa* and *S. aureus* was not linked to mechanisms directly enhancing the pathogen's iron acquisition but with mutations in global regulators, metabolism, biofilm, and motility genes. In *A. baumannii* and *K. pneumoniae*, no evidence for resistance evolution was observed. Moreover, certain pathogens lost mobile genetic elements containing virulence and antibiotic resistance genes, suggesting that pyoverdine treatment could selects for less virulent and antibiotic-sensitive strains.

## Discussion

We investigated whether pyoverdines produced by environmental *Pseudomonas* spp. have antibacterial activities against human opportunistic pathogens through iron sequestration and withholding this trace element from pathogens. Our screen involving 320 environmental isolates and 12 pathogen strains revealed five top pyoverdine candidates with broad-spectrum activity. These candidates could be distinguished from non-inhibitory pyoverdines by their high CCS values (standing for larger and more complex molecules) and their higher iron complexation stability. Experiments with crude- and HPLC-purified pyoverdines showed that pyoverdines completely stall the growth of *A. baumannii* and *S. aureus*, while showing intermediate activity against *K. pneumoniae* and *P. aeruginosa*. When administered as treatment to infected *G. mellonella* larvae, we observed significantly increased host survival rates in infections with *A. baumannii* and *K. pneumoniae*, demonstrating that pyoverdine exhibits antibacterial activity in hosts. Furthermore, we found low toxicity of pyoverdines at effective concentrations and observed low potentials for resistance evolution. Overall, our results reveal pyoverdines from non-pathogenic *Pseudomonas* spp. as potent antibacterials against several human opportunistic pathogens.

Our novel treatment approach assumes that pyoverdines sequester iron and thereby induce iron starvation and growth arrest in pathogens. Several of our findings support this view. First, the growth-inhibiting effect of our top pyoverdine candidates exclusively occurred under iron-limited conditions, while pathogen growth was restored under iron-replete conditions (*Figure 1*, *Figure 3—figure supplement 2*). Second, non-inhibitory ferri-pyoverdines were more prone toward CIU (collision induced unfolding) than inhibitory ferric-pyoverdines (*Figure 2D*), suggesting that inhibitory pyoverdines have a higher iron-binding affinity. Third, the pyoverdine precursor ferribactin did not inhibit pathogen growth (*Figure 3—figure supplement 1*). The two molecules are identical apart from one ring in the chromophore core that is not cyclized and not oxidized in ferribactin, which compromises the binding of iron. These findings suggest that pyoverdine does not target the pathogens directly, but rather indirectly via the induction of iron starvation.

We observed that pyoverdines showed pathogen-specific efficacies in curbing bacterial growth (*Figure 3*). We propose that differences can be explained by the various strategies of these pathogens to cope with iron stress. *A. baumannii* and *S. aureus* produce siderophores that have a much simpler chemical structures than pyoverdines and are expected to bind iron with lower affinity (*A. baumannii*: acinetobactin [association constant: $10^{26}$ M$^{-1}$], fimsbactin [$10^{27}$ M$^{-1}$], and baumannoferrin [unknown] [*Bohac et al., 2019*; *Shapiro and Wencewicz, 2016*; *Sheldon and Skaar, 2020*], *S. aureus*: staphyloferrin A+B [unknown] (*Hammer and Skaar, 2011*) than pyoverdine ($10^{32}$ M$^{-1}$) (*Meyer and Abdallah,*

*1978*). Hence, the growth suppression of *A. baumannii* and *S. aureus* most likely occurs because their siderophores are too weak to successfully compete with pyoverdine. The situation is different for *K. pneumoniae*, which produces enterobactin, the siderophore with the highest known iron affinity ($10^{52}$ $M^{-1}$) (*Harris et al., 1979*). Consistent with the concept of iron competition, we observed that pyoverdine treatment is less effective against *K. pneumoniae* and was only potent for one out of three pyoverdines (3G07, *Figure 3*). Finally, pyoverdine treatment is not expected to work well against *P. aeruginosa*, which itself produces a version of this siderophore. The two competing pyoverdines likely have similar iron affinities and the presence of pyoverdine receptors in the pathogen might foster the uptake of the supplemented pyoverdine. Our results indeed support the view that pyoverdine treatment is not effective against *P. aeruginosa*. Taken together, we predict that pyoverdine treatment could be highly potent against certain pathogens like *A. baumannii* and *S. aureus* that have less efficient siderophores and are unable to use the pyoverdine as iron source.

For every new treatment, it is important to consider whether it has unintentional consequences for the targeted pathogen. Because our approach does not kill pathogens directly, there are multiple ways of how pathogens may react to pyoverdine treatment. For example, pathogens could increase the production of their own siderophores in response to the severe iron limitation induced by pyoverdines. Alternatively, siderophores can serve as signaling molecules in certain species like *P. aeruginosa*, regulating the expression of additional virulence factors like proteases and toxins (*Lamont et al., 2002*). While pyoverdine treatment may affect such signaling cascades in other pathogens as well, and thus potentially increase virulence factor production, we observed the opposite in *S. aureus*, which showed a loss of virulence genes. Similarly, iron limitation can increase (*Johnson et al., 2008*) or decrease (*Banin et al., 2005*; *Lin et al., 2012*; *Kang and Kirienko, 2018*) the formation of biofilms, depending on the species or strain (*Gentile et al., 2014*; *Oliveira et al., 2021*), and it is well possible that pyoverdine treatment thus affects the propensity of pathogens to form biofilms. Finally, pyoverdines may have differential effects on the various members of the microbiota or the different pathogens in polymicrobial infections and thereby induces shifts in species composition due to the unequal suppression of community members. In complex communities, the inhibitory effect of pyoverdine on the pathogen may get diluted as the burden of increased iron limitation is shared across community members. All these considerations show that it will be crucial to assess the consequences of pyoverdine treatments in more complex settings including hosts.

Not only bacteria but also host cells need iron. Consequently, pyoverdine treatment could negatively affect host iron homeostasis. Our toxicity assays reveal that there is a range of pyoverdine concentrations for which only mild adverse effects against cell lines and *G. mellonella* larvae are observed and no hemolysis occurs. Especially the finding that pyoverdines have no negative effects on the survival of non-replicating NSC-34 cells is encouraging, as this cell line is representative of intact host tissue, in which iron is bound to strong chelators such as transferrin, lactoferrin, or ferritin (*Cassat and Skaar, 2013*; *Skaar, 2010*). Additionally, recent research has demonstrated that purified pyoverdine remains non-toxic even after 72 hr of treatment at concentrations up to 200 µM (*Kang et al., 2024*). The result that pyoverdines are unable to obtain iron from hemoglobin is encouraging, too. Previous work showed that the pyoverdine of *P. aeruginosa* can retrieve iron from human transferrin when working in concert with proteases (proteolytically degrading transferrin) (*Döring et al., 1988*) or phenazines (spontaneously reducing ferric to ferrous iron) (*Cox, 1986*). However, these *P. aeruginosa* specific mechanisms require the presence of this pathogen (*Wolz et al., 1994*), which is not the case when purified pyoverdines are used as a treatment against other pathogens like *A. baumannii*. Nonetheless, we found that pyoverdines can have certain adverse effects, especially for the rapidly proliferating HEK293 cells (*Figure 4*). These cells represent a regenerating tissue and thus have high iron demands. Similarly, we observed mild adverse effects for some pyoverdines (3G07) when administered at high concentrations to *G. mellonella* larvae (*Figure 5*). These findings corroborate the results from previous studies on *Caenorhabditis elegans*, where pyoverdines were found to interfere with host iron homeostasis (*Kang and Kirienko, 2020*; *Kirienko et al., 2013*). These considerations highlight that finding the right pyoverdine concentration will be key to meet the fine line between maximizing pathogen inhibition and minimizing tissue damage. By contrast, the adverse host effects of pyoverdines might come with additional opportunities, for instance, in cancer therapy, where siderophores reduce iron levels in tumors and slow tumor progression (*Pita-Grisanti et al., 2022*).

We propose that pyoverdine treatment could synergistically interact with the host innate immune system. Mammalian neutrophils produce siderocalin to sequester bacterial siderophore-iron complexes, rendering them non-functional (*Cassat and Skaar, 2013*). While this strategy is efficient against many siderophores, some bacteria have evolved so-called stealth siderophores that no longer bind to siderocalin. Pyoverdines are such stealth siderophores (*Peek et al., 2012*). Regarding pyoverdine treatment, this means that pathogens with regular siderophores will hit a double wall: their siderophores are immobilized by siderocalins, while pyoverdines lock away the remaining iron.

Another promising finding of our work is that the potential for resistance evolution against pyoverdines seems low. Conceptually, one reason for low-resistance evolution could be that pyoverdines, unlike many antibiotics, are not internalized into bacterial cells. Thus, classic resistance mechanisms such as reduced drug influx, increased drug efflux, and intracellular target modification cannot operate (*Rezzoagli et al., 2018*). We initially expected that resistance evolution would involve mechanisms that improve the pathogen's iron acquisition. This could include the upregulation of the pathogen's own siderophores or switching from siderophore-based ferric iron acquisition to the use of reductases that foster ferrous iron uptake (*Rezzoagli et al., 2018*). However, our sequencing analysis did not reveal mutations in genes associated with iron uptake systems. One explanation is that such mutations (albeit beneficial) might not reach high frequencies in populations because increased siderophore production and secretion would increase the iron acquisition of resistant and non-resistant cells alike (*Rezzoagli et al., 2018*). We instead observed an overrepresentation of mutations in regulatory, biofilm and motility, and metabolic genes. While we do not know the phenotypes of these mutations, one possibility is that increased biofilm formation could offer some protection against pyoverdine treatment, for example, by restricting pyoverdine from chelating free iron. Important to note is also that patterns of resistance evolution may differ between in vitro and in vivo settings (*Bell and MacLean, 2018*; *Sommer et al., 2017*). For example, our experimental evolution setup did not allow for horizontal transfer of siderophore receptors, a scenario that might occur in the context of polymicrobial infections and confer resistance to pyoverdine therapy.

In conclusion, our treatment approach using pyoverdine is based on the principle of paying like with like. Pathogens secrete siderophores to scavenge iron for their growth. We add a strong heterologous siderophore as treatment to withhold iron from pathogens to curb their growth. The concept of targeting bacterial infections with both natural and synthetic iron-chelators has been highlighted in the literature for quite some time (*Qiu et al., 2011*; *Kontoghiorghes et al., 2010*; *Ribeiro et al., 2022*; *Coraça-Huber et al., 2018*; *Hatcher et al., 2009*). For instance, the siderophore desferoxamine (produced by *Streptomyces* species) is used in clinical settings to treat iron overload (*Mobarra et al., 2016*). However, its efficacy as antibacterial is limited because of its low iron affinity ($10^{27}$ M$^{-1}$, compared to pyoverdine and enterobactin) (*Elalfy et al., 2023*) and several pathogens have receptors for deferoxamine uptake and can thus use it as an iron source (*Eto et al., 2013*). Synthetic iron chelators like deferiprone and deferasirox have been proposed as alternatives to overcome these limitations. However, the minimum inhibitory concentration (MIC) of deferiprone needed to inhibit *P. aeruginosa* was found to be up to 36.7 times higher than the plasma levels in human therapy, raising concerns about its cytotoxicity and clinical viability for treating bacterial infections (*Thompson et al., 2012*; *Visca et al., 2013*). Furthermore, at sub-MIC levels, deferiprone only partially suppressed the growth of *P. aeruginosa* and *A. baumannii,* indicating insufficient iron sequestration from these pathogens (*Visca et al., 2013*; *de Léséleuc et al., 2012*). Our approach offers a distinct advantage by utilizing naturally evolved iron chelators from non-pathogenic species, which possess unique chemical structures and exceptionally high iron affinities. These properties reduce the likelihood of their exploitation as iron sources by pathogens and provide a competitive advantage over siderophores with lower iron affinities. Combined with low toxicity to hosts at therapeutically effective concentrations and minimal potential for resistance evolution, these natural chelators hold promise as a new class of effective antibacterials, either as standalone treatments or in combination with conventional antibiotics.

## Materials and methods

### Bacterial strains

We used a strain collection of 320 natural isolates, originating from soil and pond habitats. The isolates originate from 16 independent soil and water samples (20 isolates per sample) collected on the Campus Irchel Park of the University of Zurich. In previous work, we have extensively studied and characterized all isolates (*Butaitė et al., 2017*; *Butaitė et al., 2018*; *Butaitė et al., 2021*; *Kramer et al., 2020*). Particularly, we used 315 isolates where sequencing of the rpoD housekeeping gene confirmed that they belong to the group of fluorescent pseudomonads that are non-pathogenic to humans (*Butaitė et al., 2017*). For the remaining five isolates, the rpoD gene could not be amplified, yet they were still included in our study. Moreover, siderophore screens previously revealed that a large proportion of the isolates can produce and secrete pyoverdines (*Butaitė et al., 2017*), the high-iron affinity siderophores we focus on in our study. The exact sampling and isolation procedures together with the pyoverdine production profiles are described elsewhere (*Butaitė et al., 2017*; *Kramer et al., 2020*). All bacterial natural isolates and pathogenic species used in our study will be available from the authors upon request.

For the supernatant screening assay, we used a collection of 12 strains of opportunistic human pathogens (*Table 1*), including *A. johnsonii*, *A. junii*, *C. sakazakii*, *K. michiganensis* (belonging to the *K. oxytoca* complex) (*Yang et al., 2022*), and *Shigella* sp. (all provided by the laboratory of Leo Eberl, University of Zurich), and *B. cenocepacia* strains H111 and K56-2, *P. aeruginosa* strains PA14 and PAO1 (ATCC 15692), *E. coli* K12 and *S. aureus* strains Cowan and JE2 (NARSA) from our own strain collection. For dose–response and infection experiments, we used *A. baumannii* (DSM 30007) and *K. pneumoniae* (DSM 30104), both purchased from the German Collection of Microorganisms and Cell Cultures GmbH, in addition to *P. aeruginosa* PAO1 and *S. aureus* JE2.

### Culturing conditions

Pathogen overnight cultures were grown in either 8 mL tryptic soy broth (TSB; for the two *S. aureus* strains) or 8 mL lysogeny broth (LB; all other pathogens) in 50 mL tubes at 37°C and 220 rpm agitation. Following two washing steps with 0.8% NaCl, we adjusted the overnight cultures to an optical density at 600 nm ($OD_{600}$) of 1. For overnights of environmental isolates, we followed the same procedure when working with low sample sizes, with the only difference being that culturing occurred at 28°C. For the large-scale screening experiments, we grew environmental isolates in 200 µL LB in 96-well plates shaken at 170 rpm. Cultures from plates were then directly used for experiments. For all main experiments (screen for bioactive pyoverdines and dose–response curves), we used CAA medium (1% casamino acids, 5 mM $K_2HPO_4 * 3H_2O$, 1 mM $MgSO_4 * 7 H_2O$, 25 mM HEPES buffer). This medium contains low levels of iron, triggering pyoverdine production (*Cunrath et al., 2016*).

### Supernatant screening assays

We created sterile supernatants from all the 320 *Pseudomonas* isolates. Specifically, we transferred 2 µL of overnight cultures (grown in 96-well plates in LB in fourfold replication for 24 hr as described above) to new plates containing 200 µL of CAA medium supplemented with 250 µM 2,2'-bipyridyl. Cultures were incubated for 24 hr and subsequently centrifuged at 2250 × *g* for 10 min. The supernatants were transferred to 0.2 µm membrane filter plates (PALL AcroPrep Advance), centrifuged a second time, and frozen at –20°C.

For the supernatant screening assay, the pathogens were cultured and diluted as described above and added to 140 µL CAA medium in 96-well plates. We added 60 µL of thawed *Pseudomonas* supernatants (30%) or 0.8% NaCl as control and incubated the plates at 37°C and 170 rpm. Growth was measured as $OD_{600}$ after 24 hr using a Tecan Infinite M-200 plate reader (Tecan Group, Männedorf, Switzerland). We then scaled the growth of pathogens in the supernatant relative to their growth in the control treatment. Relative growth values smaller and larger than one indicate growth inhibition and promotion by the supernatant, respectively.

To test whether pyoverdine is responsible for pathogen growth reduction, we repeated the screening assay but supplemented the CAA medium containing the *Pseudomonas* supernatants with 40 µM $FeCl_3$. In this iron-rich condition, pyoverdine should not be able to limit iron availability and thus pathogen growth should be restored.

## Structure elucidation of pyoverdine and analyses of chemical properties

From the seven top supernatant candidates that inhibited all 12 pathogens, we extracted the pyoverdines and elucidated their structure. For this purpose, we have developed a new protocol that allows structure elucidation from low-volume supernatants using UHPLC-HR-MS/MS. The methodological details are described in *Rehm et al., 2022* and *Rehm et al., 2023*. Afterward, we assessed whether growth-inhibiting pyoverdines can be distinguished from non-inhibiting pyoverdines by their chemical properties. We focussed on polarity, CCS values, and stability upon CIU or fragmentation. To analyze chemical properties of pyoverdines, we first purified them from the supernatants using solid-phase extraction. Next, we ran the samples through an HSS C18 column of the Vanquish UHPLC system (Thermo Fisher Scientific, Waltham, MA) using formic acid (0.1%) as an eluent additive and MeCN as the organic phase and recorded the RTs (as a measure for polarity). We also measured the CCS values of pyoverdines and ferri-pyoverdines on a trapped ion mobility spectrometry (TIMS)-TOF-MS instrument (timsTOF Pro, Bruker, Bremen, Germany) at a Δ6 voltage of 100 V. Finally, we assessed the stability of ferri-pyoverdines with two approaches. First, we investigated the proneness of ferri-pyoverdines to undergo CIU. For this, we determined the difference in CCS values of the ferri-pyoverdines (ΔCCS) at a Δ6 voltage of 50 V (no structural unfolding is stimulated) and of 150 V (structural unfolding takes place depending on the stability of the iron complex). Second, we measured the NCE necessary to fragment 50% of ferri-pyoverdine ions (CE50). This experiment was conducted on a Q Exactive hybrid quadrupole-Orbitrap mass spectrometer (Thermo Fisher Scientific). Ferri-pyoverdine ions were isolated and fragmented at increasing NCE using parallel reaction monitoring, while the ion intensity of the intact ion precursor was evaluated.

## Crude pyoverdine purification

To test the efficacy of the pyoverdines against the four focal human opportunistic pathogens (*A. baumannii*, *K. pneumoniae*, *P. aeruginosa* [PAO1], and *S. aureus* [JE2]), and to assess their cytotoxicity against human cell lines, we crude-purified three pyoverdines (3A06, 3G07, s3b09). The three pyoverdines were among the five most potent ones identified in our supernatant screening assay. We adapted the method for pyoverdine purification from previous studies (*Butaitè et al., 2017*; *Meyer et al., 1997*). To stimulate pyoverdine production, 500 mL CAA medium was supplemented with 250 µM of the synthetic iron chelator 2,2'-bipyridyl to create an iron-limited environment and inoculated with 2 mL of washed overnight *Pseudomonas* cultures. We incubated the cultures at 28°C for 120 hr with agitation (170 rpm). Cultures were then centrifuged at 15,049 × *g* for 15 min. We then harvested the supernatant and adjusted its pH to 6 using 1 M HCl. We ran the supernatants over Amberlite XAD-16N resin (Sigma-Aldrich, Switzerland) columns at a rate of two drops per second and cleared the column of any salts using 500 mL Milli-Q water. Pyoverdines were then eluted with 50% methanol and fractions of 50 mL were collected. Fractions containing the highest amount of pyoverdine (measured by fluorescence, excitation: 400 nm and emission: 460 nm), were pooled. After evaporation of the methanol, we lyophilized the samples for 24 hr and stored them at –20°C. As the crude purified extracts may contain small impurities, the exact pyoverdine concentration is unknown. For the figures, concentrations were divided by 6 mg/mL (highest concentration used) and expressed as relative concentrations.

## HPLC purification of pyoverdine and ferribactin

To further validate that pyoverdines are the compounds that induce pathogen growth inhibition, we purified the most common pyoverdine (s3b09) among the most potent pyoverdines from our screens using reversed-phase HPLC. For this, we first crudely purified the pyoverdine from 9 L supernatant as described above. The crude extract was then dissolved in water and fractionated by preparative HPLC at room temperature (Agilent 1260 Infinity System, equipped with a Phenomenex Luna 5 µm C18 21.2 × 250 mm column). Ultraviolet (UV) detection was carried out at $\lambda$ = 210 nm, 230 nm, 280 nm, 350 nm, 400 nm, and 460 nm. Deionized water (Milli-Q, Millipore) (solvent A) and acetonitrile (solvent B) were used as the mobile phase with a flow rate of 15 mL/min. We eluted with a gradient of 5–100% solvent B in 40 min, followed by isocratic conditions at 100% solvent B for 10 min. The fractions containing pyoverdine were identified based on the HR-ESI-MS data acquired on a Thermo Scientific Q Exactive hybrid quadrupole-orbitrap mass spectrometer (scan range 100–2500 *m/z*, capillary voltage 4500 V, dry temperature 200°C) coupled to an UltiMate 3000 UHPLC system

(Dionex, equipped with a Kinetex 2.6 µm XB-C18 150 × 4.6 mm column; solvent A: deionized water; solvent B: acetonitrile; gradient, 5% B for 30 s increasing to 100% B in 15.5 min and then maintaining 100% B for 5 min; flow rate 0.6 mL/minute; UV–Vis detection 200–600 nm). The fractions containing pyoverdine eluting at a RT of 3.84 min in the LC-MS chromatogram (*Figure 3—figure supplement 3*) were further purified on a semi-preparative reversed-phase HPLC at room temperature (Agilent 1260 Infinity System, equipped with a Phenomenex Luna 5 µm Phenyl-Hexyl 10 × 250 mm column, $\lambda$ = 210 nm, 230 nm, 280 nm, 350 nm, 400 nm, and 460 nm monitored by UV detection). The elution gradient was increased from 5% to 15% solvent B for 30 min followed by a gradient shift from 15% to 100% in 5 min, and finally isocratic condition at 100% solvent B for 5 min. During the pyoverdine purification, we noted high amounts of the precursor molecule ferribactin (RT of 2.42 min in the LC-MS chromatogram; *Figure 3—figure supplement 4*). We thus purified it alongside pyoverdine using the same method.

## Dose–response curves

To determine the inhibition potential of crude- and HPLC-purified pyoverdines, as well as ferribactin, we subjected *A. baumannii. K. pneumoniae, P. aeruginosa,* and *S. aureus* to serially diluted extracts. For this, the extracts were dissolved in CAA medium, filter sterilized, and diluted in the same medium. Bacterial overnight cultures were grown and prepared as described above and added to a total of 200 µL medium on a 96-well plate with four replicates per extract dilution. As positive control where growth inhibition is expected, we subjected all three pathogens to a dilution series of the antibiotic ciprofloxacin (highest concentration 4 µg/mL) in the exact same way as for the extracts. We incubated the plates at 37°C statically in the plate reader and measured the $OD_{600}$ every 15 min for 24 hr. Before each measurement, plates were shaken for 15 s. We repeated the experiment up to five times with different initial extract starting concentrations. This was done to ensure that the full dose response to pyoverdine was captured. Thus, sample size per individual extract concentration varied between 4 and 20 replicates. We subtracted the blank values and the background values caused by the treatment of the respective extract concentration from the measured growth data and calculated the integral (area under the growth curve) using the R package Growthcurver (*Sprouffske and Wagner, 2016*). We then expressed growth relative to the control medium, where bacteria grew in CAA without extract or antibiotic addition. Finally, we plotted extract/antibiotic concentration vs. relative growth and fitted dose–response curves (see below).

## Cytotoxicity assays

To determine the cytotoxicity of pyoverdine treatments against mammalian cells, we subjected human embryonic kidney 293 (HEK-293; Invitrogen R78007, authenticated) and mouse motor neuron-like neuroblastoma-spinal cord hybrid (NSC-34; Cedarlane CLU140, authenticated) cells to the crude extract of the three most potent pyoverdines (3A06, 3G07, s3b09). HEK-293 cells were cultured in Dulbecco's modified Eagle medium (DMEM; Sigma, D5671) supplemented with 10% fetal bovine serum (FBS; Gibco, 10270-106), 1× GlutaMAX (Gibco, 35050-061), 100 U/mL penicillin and 100 µg/mL streptomycin (Gibco, 15140-122). NSC-34 cells were proliferated on Matrigel (Corning, 354234)-coated dishes in DMEM supplemented with 10% FBS, 1× GlutaMAX, 100 U/mL penicillin, and 100 µg/mL streptomycin. For experiments, differentiation was induced by switching to DMEM/F12 medium (Gibco, 21331-020) supplemented with 1× GlutaMAX, 1× B27+ supplement (Gibco, 17504-044), 1× N2 supplement (Gibco, 17502-048), 20 ng/mL BDNF (PeproTech, 450-02), 20 ng/mL GDNF (Peptro-Tech, 450-10), 100 U/mL penicillin, and 100 µg/mL streptomycin. All cells were cultured at 37°C with saturated humidity and an atmosphere of 5% $CO_2$. Cells were routinely tested for mycoplasma contamination every 1–2 months using a PCR detection kit (Sigma-Aldrich, MP0035).

For the assay, 10,500 cells/well were plated on 96-well plates (Greiner Bio-One, 655090) and, after 48 hr, the culture medium was replaced and the pyoverdines were added at the appropriate concentrations. For this, pyoverdine stocks at 240 mg/mL were diluted in Milli-Q water such that the addition of 5 µL of the pyoverdine solution to 195 µL of the culture medium resulted in final relative concentrations between 1 and 0.016. As the exact pyoverdine concentration in the crude extracts is unknown, the above concentrations were divided by the highest concentration used (6 mg/mL) and expressed as relative concentrations in the figure. After further 48 hr, the pyoverdine-containing culture medium was removed, and cells were incubated with 1 µg/mL thiazolyl blue tetrazolium bromide (Sigma,

D5655) in 100 µL of fresh medium for 1.5 hr (NSC-34) or 40 min (HEK-293). The reaction was stopped, and formazan crystals simultaneously solubilized with the addition of 100 µL of a solution containing 10% sodium dodecyl sulfate (Sigma-Aldrich, 05030) and 0.03% HCl (Supelco, 100319). Finally, the absorbance of cell debris and other contaminants at 630 nm were subtracted from the absorbance of the solubilized formazan at 570 nm.

## Hemolysis assay

To determine the hemolytic activity of pyoverdines, we quantified sheep erythrocyte hemolysis at different pyoverdine concentrations. We first centrifuged 25 mL of fresh sheep blood (Chemie Brunschwig AG, Switzerland) in a 50 mL tube at 1500 × $g$ for 5 min at 4°C. Plasma was then gently aspirated, replaced by PBS, and tubes were gently inverted to mix. This washing procedure was repeated three times. Pyoverdines were diluted in PBS to final relative concentrations of 1, 0.5, 0.25, 0.2, 0.02, and 0.002. PBS and 0.1% Triton X-100 (Sigma-Aldrich) served as negative and positive control, respectively. 100 µL erythrocyte solution was mixed with either 100 µL pyoverdine solution or the controls in a 96-well plate in triplicates. Following an incubation at 37°C and 120 rpm for 1 hr, the plate was centrifuged at 1036 × $g$ for 5 min at 4°C. Subsequently, 100 µL of the supernatant were transferred to a new 96-well plate. Hemoglobin release was measured at 570 nm in the plate reader and the percent hemolysis was calculated relative to the positive control [% hemolysis = (OD570 of sample – negative control)/(positive control – negative control)].

## Infection model

We tested the efficacy of pyoverdine to treat infections in larvae of the host model *G. mellonella*. All larvae were ordered from a local vendor (Bait Express GmbH, Basel, Switzerland) and stored without food at 16°C in the dark for up to 3 days. In a first step, we determined the mean weight of 336 final instar *G. mellonella* larvae (mean = 435 mg). For all subsequent experiments, we weighed the larvae and only used individuals with a mean weight ±20% (range: 340–530 mg). Larvae were immobilized on ice for 15 min, surface-sterilized with 70% ethanol, and randomly distributed to individual wells on 24-well plates. Ten larvae were used per infection–treatment combination. To inject bacteria and administer treatments, we used sterile needles (Sterican 26G, 0.45 × 12 mm [Braun]) and syringes (1 mL injekt-F [Braun]) attached to a syringe pump (New Era Pump Systems Inc, model NE-300) set to a flow rate of 5 mL/hr. For both infection and treatment delivery, a volume of 10 µL per larvae was injected between the posterior larval prolegs.

On the pathogen side, we determined the LD$_{50}$ (lethal dose, killing 50% of the larvae) for *A. baumannii* (1.8 * 10$^5$ CFU/larvae) and *K. pneumoniae* (8.9 * 10$^5$ CFU/larvae) (*Figure 5—figure supplement 1*). For *P. aeruginosa*, we use 56 CFU/larvae because all doses tested killed 100% of the larvae (*Figure 5—figure supplement 1*). As negative control, we administered 10 µL/larvae of sterile PBS. Following infections, larvae were individually placed in wells of 24-well plates and incubated at 37°C in the dark for 4 hr.

On the treatment side, we used pyoverdines 3A06, 3G07, and s3b09 at three concentrations that were non-toxic for the NSC-34 cells. For this, we prepared stocks of 60 mg/mL, 30 mg/mL, and 6 mg/mL of crude pyoverdines. Since the injected treatment volume was 10 µL per larvae, the final pyoverdine concentrations were 0.6 mg, 0.3 mg, and 0.06 mg per larvae, respectively. When scaling theses concentrations to the doses used for the in vivo dose–response curves (*Figure 4*), the relative pyoverdine concentrations were 0.1, 0.05, and 0.01. Pyoverdine treatments were administered 4 hr after the infection through injections. Prior to treatment, larvae were immobilized on ice for 15 min. We also applied treatments to larvae infected with sterile PBS to quantify potential negative effects of the treatment in the absence of an infection. Following treatment, larvae were put back to their allocated well of the 24-well plate and kept at 37°C in the dark.

From the 12th hour post infection (hpi) onward, we regularly checked the survival of all larvae for a total of 48 hpi. Specifically, larvae were poked with a pipette tip and individuals not responding to the physical stimulation were considered dead. We conducted three independent experiments per pathogen–treatment combination with 10 larvae per experiment, resulting in a total of 30 larvae per pathogen–treatment combination. Larvae in their last instar stage were purchased from a local vendor (Bait Express GmbH), and different batches of larvae were used for the three independent experiments.

## Experimental evolution experiment plate design and setup

We first determined the IC50 concentrations of (crude) pyoverdines 3A06, 3G07, and s3b09, and the antibiotic ciprofloxacin against all four pathogens. For this, we fitted logistic regressions to the dose–response curves. We used these IC50 values as treatment concentrations during experimental evolution (*Supplementary file 1b*). We also mixed different pyoverdines in double and triple combo treatments. In combo treatments, pyoverdine concentrations were equivalent to those of the single treatments, meaning that we divided the single-treatment concentrations by two or three. The experimental evolution experiment included (i) three single pyoverdine treatments (3A06, 3G07, s3b09), (ii) three double pyoverdine treatments (3A06 and 3G07, 3A06 and s3b09, 3G07, and s3b09), (iii) one triple pyoverdine treatment (3A06, 3G07, and s3b09), and (iv) one ciprofloxacin treatment for *A. baumannii*, *P. aeruginosa*, and *S. aureus*. For *K. pneumoniae*, which was only affected by one pyoverdine (3G07), the experiment consisted of one single pyoverdine treatment and one ciprofloxacin treatment.

For each pathogen–treatment combination, we included six independently replicated lineages (populations) on 96-well plates. Each plate further contained two control populations that evolved in plain CAA medium, except for *K. pneumoniae*, for which all six controls were on the same plate. The six replicate populations were arranged along the diagonal of 96-well plates or at the border of the plates, in such a way that they were surrounded by medium on all sides. This plate layout minimized the chance of cross-contamination and resulted in a total of 10 plates (three each for *A. baumannii*, *P. aeruginosa*, and *S. aureus* and one for *K. pneumoniae*).

Prior experimental evolution, we prepared batches of media for the entire duration of the experiment. Specifically, all treatments were prepared in CAA medium, then aliquoted into the respective wells of 96-well plates, and all plates were stored at –20°C. At the start, we grew the pathogens overnight, washed and diluted them as described in the 'Materials and methods', and transferred 2 µL of the diluted pathogens into 200 µL medium in 96-well plates. Subsequently, the plates were incubated at 37°C for 22 hr, shaken at 170 rpm. After incubation, we measured growth (OD$_{600}$) of all evolving lineages with the plate reader. We then diluted the evolving cultures 1:1000 in CAA medium and added them to a fresh treatment plate thawed from the freezer. After the transfer, the new plates were again incubated at 37°C and 170 rpm agitation for 22 hr. We added glycerol to the old plates to a final concentration of 25% and stored them at –80°C. We observed a steady decrease in growth in all *S. aureus* populations, suggesting that we overdiluted these cultures. Consequently, we reduced the dilution to 1:10 from day 4 onward. Conversely, we observed a steady increase in growth for *A. baumannii* and *P. aeruginosa* populations, suggesting that these cultures were not diluted enough. Consequently, we increased dilution rates to appropriate levels on days 4, 8, and 12.

## Phenotypic screening of evolved populations

We screened evolved populations for improved growth performance relative to the ancestor under pyoverdine treatment, which we took as a proxy for resistance evolution. For this purpose, we thawed the final plates (day 16) of the experimental evolution experiment and transferred 2 µL of all populations to 200 µL of LB or TSB (for *S. aureus*) to create overnight cultures in 96-well plates. We further included ancestor cultures of all four species. After incubation for 18 hr at 37°C and 170 rpm, we diluted overnights 1:1000 with 0.8% NaCl and subjected all evolved populations to the conditions they had evolved in. Ancestors were simultaneously exposed to the respective treatment conditions. The culturing volume and conditions matched those used during experimental evolution, but this time we incubated the plates in a plate reader, measuring OD$_{600}$ every 15 min to obtain growth kinetics. The population screen was repeated twice independently. From the growth kinetics, we first calculated the area under the curve (AUC) for each growth curve. AUC values were then scaled relative to the growth of the ancestor in the untreated CAA control medium. For each evolved population, we then calculated the difference in growth in comparison to the mean growth of the ancestor subjected to the same treatment.

## Phenotypic screening of evolved clones

To screen for resistant clones within populations, we selected the pathogen populations with the lowest and highest growth difference for each single treatment and for one combination treatment and picked eight random colonies. We transferred them to 200 µL LB or TSB (*S. aureus*) in 96-well

plates and incubated plates for 18 hr at 37°C (170 rpm) to create overnight cultures. We followed the same protocol as above for quantifying the difference in growth of evolved and ancestral clones.

## Genomic analyses of evolved populations and clones

We sequenced the genomes of the four ancestors, four control populations that had evolved in medium without pyoverdine treatment, and two random clones (out of the eight picked) per population that had received pyoverdine treatment and were used for clonal growth analysis (*Figure 6B*). For sequencing, we grew the clones and populations in 12 mL LB or TSB (*S. aureus*) at 37°C and 170 rpm and harvested cultures upon reaching an $OD_{600}$ between 0.8 and 1 by centrifugation (7500 rcf, 3 min). We washed cultures in 0.8% NaCl and resuspended the pellet in DNA shield buffer (Zymo Research). Samples were then sent to MicrobesNG (Birmingham, UK) for library preparation and sequencing. Whole-genome sequencing was performed on the Illumina NovaSeq6000 platform (Illumina, San Diego) using a 250 bp paired end protocol. Adapter sequences were trimmed using Trimmomatic v0.30 (*Bolger et al., 2014*) with quality cut-off of Q15. De novo assemblies were performed using SPAdes v3.7 (*Bankevich et al., 2012*) and contigs annotated with Prokka v1.11 (*Seemann, 2014*).

Variants were predicted relative to the reference genomes ATCC 19606 (*A. baumannii*), ATCC 13883 (*K. pneumonia*), ATCC 15692 (*P. aeruginosa*), and CP020619.1 (*S. aureus*) using the breseq pipeline (*Deatherage and Barrick, 2014*). In the pipeline, we specified that medium-adapted controls were treated as populations, whereas the treatment-adapted colonies picked for sequencing were treated as clones.

## Data analyses

All statistical analyses were performed in R 4.0.2 and RStudio version 1.3.1056. We conducted Shapiro–Wilk tests and consulted diagnostic plots to check whether our data (residuals) were normally distributed. For normally distributed data sets, we used linear models and two-sided tests for all analyses. For non-normally distributed data sets, we used nonparametric rank tests. We used ANOVA to test whether chemical properties differ between growth-inhibiting and non-inhibiting pyoverdines, in their ferri- versus apo-pyoverdine state. For the in vitro dose–response and cytotoxicity experiments, we fitted dose–response curves with either four- or five-parameter logistic regressions using the nplr package (*Commo and Bot, 2016*). Survival analyses were performed using the Cox proportional hazards regression model or the log-rank test (adjusted for multiple comparisons using the Holm method) from the survival package (*Therneau, 2022*). We used two-sided t-tests, Welch's t-tests, or Wilcoxon tests to compare growth differences between the ancestor and the evolved populations under treatment. To compare growth differences between the ancestor and the evolved clones under treatment, we used one- and two-way ANOVAs. To construct the cladogram in *Figure 1—figure supplement 2*, we followed the procedure described in *Kramer et al., 2020* using partial *rpoD* gene sequences with lengths ≥600 bp. We used iTOL tool (*Letunic and Bork, 2016*) for the graphic presentation of the cladogram and to link phylogeny to the supernatant effects.

## Acknowledgements

We thank Richard Allen, Dominik Bär, Johanna Giger, Kevin Schiefelbein and Julien Weber for laboratory assistance, and Markus Seeger for insightful discussion. This work was supported by funding from the Swiss National Science Foundation (grant numbers: 31003A_182499 to RK, PP00P3_144862 and PP00P3_176966 to MP) and a Candoc Grant (Forschungskredit) from the University of Zurich (to MPB).

## Additional information

### Funding

| Funder | Grant reference number | Author |
| --- | --- | --- |
| Swiss National Science Foundation | 31003A_182499 | Rolf Kümmerli |

| Funder | Grant reference number | Author |
|---|---|---|
| Swiss National Science Foundation | PP00P3_144862 | Magdalini Polymenidou |
| Swiss National Science Foundation | PP00P3_176966 | Magdalini Polymenidou |
| University of Zurich | Candoc | Manuela Pérez-Berlanga |

The funders had no role in study design, data collection and interpretation, or the decision to submit the work for publication.

## Author contributions

Vera Vollenweider, Conceptualization, Resources, Data curation, Software, Formal analysis, Supervision, Validation, Investigation, Visualization, Methodology, Writing – original draft, Writing – review and editing; Karoline Rehm, Manuela Pérez-Berlanga, Data curation, Formal analysis, Methodology, Writing – original draft; Clara Chepkirui, Data curation, Formal analysis, Supervision, Methodology; Magdalini Polymenidou, Supervision, Funding acquisition, Project administration; Jörn Piel, Laurent Bigler, Supervision, Funding acquisition, Writing – original draft, Project administration; Rolf Kümmerli, Conceptualization, Resources, Data curation, Formal analysis, Supervision, Funding acquisition, Validation, Investigation, Methodology, Writing – original draft, Project administration, Writing – review and editing

## Author ORCIDs

Vera Vollenweider ⬤ https://orcid.org/0000-0003-2920-3031
Manuela Pérez-Berlanga ⬤ https://orcid.org/0000-0001-9064-9724
Magdalini Polymenidou ⬤ https://orcid.org/0000-0003-1271-9445
Jörn Piel ⬤ https://orcid.org/0000-0002-2282-8154
Laurent Bigler ⬤ https://orcid.org/0000-0003-3548-3594
Rolf Kümmerli ⬤ https://orcid.org/0000-0003-4084-6679

Reviewer #1 (Public review): https://doi.org/10.7554/eLife.92493.3.sa1
Reviewer #2 (Public review): https://doi.org/10.7554/eLife.92493.3.sa2
Author response https://doi.org/10.7554/eLife.92493.3.sa3

# Additional files

## Supplementary files

• Supplementary file 1. Supplementary tables a-e.

• MDAR checklist

## Data availability

The raw data underlying the figures are available from the Figshare depository (https://doi.org/10.6084/m9.figshare.26388565). The sequencing data for this study have been deposited in the European Nucleotide Archive (ENA) at EMBL-EBI under accession number PRJEB78468 (https://www.ebi.ac.uk/ena/browser/view/PRJEB78468).

The following datasets were generated:

| Author(s) | Year | Dataset title | Dataset URL | Database and Identifier |
|---|---|---|---|---|
| Vollenweider V, Rehm K, Chepkirui C, Pérez-Berlanga M, Polymenidou M, Piel J, Bigler L, Kümmerli R | 2024 | Vollenweider et al (2024) eLife Raw Data | https://doi.org/10.6084/m9.figshare.26388565 | figshare, 10.6084/m9.figshare.26388565 |

*Continued on next page*

*Continued*

| Author(s) | Year | Dataset title | Dataset URL | Database and Identifier |
|---|---|---|---|---|
| Vollenweider V, Rehm K, Chepkirui C, Pérez-Berlanga M, Polymenidou M, Piel J, Bigler L, Kümmerli R | 2024 | Pyoverdines against pathogens | https://www.ebi.ac.uk/ena/browser/view/PRJEB78468 | European Nucleotide Archive, PRJEB78468 |

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
