## [Editor Report · eLife Assessment]

This **important** study highlights the use of siderophores as antibacterials, and the authors also discuss the consequences and efficacy of 'siderophore therapy' in more complex communities/environments. The evidence supporting the overall hypotheses ranges is largely **convincing**. The work will be of broad interest to people working in the fields of evolutionary ecology, microbiology and medical sciences.

---

## [Referee Report · Reviewer #1 (Public review)]

Summary:

In an era of increasing antibiotic resistance, there is a pressing need for the development of novel sustainable therapies to tackle problematic pathogens. In this study, the authors hypothesize that pyoverdines - metal-chelating compounds produced by fluorescent pseudomonads - can act as antibacterials by locking away iron, thereby arresting pathogen growth. Using biochemical, growth and virulence assays on 12 opportunistic pathogens strains, the authors demonstrate that pyoverdines induce iron starvation, but this affect was highly context dependent. This same effect has been demonstrated for plant pathogens, but not for human opportunistic pathogens exposed to natural siderophores. Only those pathogens lacking (1) a matching receptor to take up pyoverdine-bound iron and/or (2) the ability to produce strong iron chelators themselves experienced strong growth arrest. This would suggest that pyoverdines might not be effective against all pathogens, thereby potentially limiting the utility of pyoverdines as global antibacterials.

Strengths:

The work addresses an important and timely question - can pyoverdines be used as an alternative strategy to deal with opportunistic pathogens? In general, the work is well conducted with rigorous biochemical, growth and virulence assays. In line, the work is clearly written, and the findings are supported by high-quality figures.

Weaknesses:

I do not think there are any 'weaknesses' as such. The authors have taken all suggestions on board and this has greatly improved the quality and robustness of the work

---

## [Referee Report · Reviewer #2 (Public review)]

In this work, Vollenweider et al. examine the effectiveness of using natural products, specifically molecules that chelate iron, to treat infectious agents. Through the purification of 320 environmental isolates, 25 potential candidates were identified based on inhibition assays and further screened. The structural information and chemical composition of these candidates were determined. Using a series of well-described and standard assays, the authors show that three compounds have some effect in reducing mortality in a simple in vivo model.

The paper is well-structured and thorough; targeting virulence factors in this manner is an excellent approach. However, my enthusiasm is dampened by the mediocre effects of the compounds. A reduction in the hazard ratio is reported, indicating that the compounds are having an effect, but without comparison to other iron-chelating molecules or current standards of care, it is difficult to contextualize the significance of these reductions.

I am less convinced by a claim from the abstract: "Furthermore, experimental evolution combined with whole-genome sequencing revealed reduced potentials for resistance evolution compared to an antibiotic." Perhaps this is a semantic issue, but what is meant by "potential for resistance evolution"? My understanding is that this refers to mutations or sets of mutations that would be favored under selective pressure, allowing the bacteria to more easily climb a fitness landscape peak. However, the authors present a different result: the bacteria did not grow better after selection in different conditions (except for the positive control using ciprofloxacin). They correctly suggest that there may be individuals in the populations that have developed resistance and recommend isolating 8 from each treatment for testing. However, they then use the mean value of these individuals to conclude that there is no difference from the ancestor. This seems incorrect-surely the point of using individuals is not to compare them as a group but to determine if any one has a growth rate outside the expected distribution. In short, Figure S10 does not seem to support the findings reported in line 417.

A final consideration for the evolution experiment is the choice of a bactericidal antibiotic. It might have been more appropriate to use a bacteriostatic drug as a control. However, I feel that additional work on this topic is beyond the scope of the current paper.

Similarly, it would be interesting to consider how evolving the isolates in iron-limited media would affect resistance levels. Currently, I think the difference in growth rate is attributed to the iron-scavenging nature of the siderophores. In future work, this could be tested, and an evolution experiment in which iron availability is measured could provide valuable insights. To clarify, I believe this work is not necessary for the current paper, but it would be an interesting avenue for future research.

---

## [Author Response]

The following is the authors’ response to the original reviews.

**Reviewer #1 (Public Review):**
Summary:In an era of increasing antibiotic resistance, there is a pressing need for the development of novel sustainable therapies to tackle problematic pathogens. In this study, the authors hypothesize that pyoverdines - metal-chelating compounds produced by fluorescent pseudomonads - can act as antibacterials by locking away iron, thereby arresting pathogen growth. Using biochemical, growth, and virulence assays on 12 opportunistic pathogens strains, the authors demonstrate that pyoverdines induce iron starvation, but this effect was highly context-dependent. This same effect has been demonstrated for plant pathogens, but not for human opportunistic pathogens exposed to natural siderophores. Only those pathogens lacking (1) a matching receptor to take up pyoverdine-bound iron and/or (2) the ability to produce strong iron chelators themselves experienced strong growth arrest. This would suggest that pyoverdines might not be effective against all pathogens, thereby potentially limiting the utility of pyoverdines as global antibacterials.Strengths:The work addresses an important and timely question - can pyoverdines be used as an alternative strategy to deal with opportunistic pathogens? In general, the work is well conducted with rigorous biochemical, growth, and virulence assays. The work is clearly written and the findings are supported by high-quality figures.Weaknesses:I do not think there are any 'weaknesses' as such. However, it is well known that siderophore production is highly plastic, typically being upregulated in response to metal limitation (as well as toxic metal stress). Did the authors quantify whether pyoverdine supplementation altered siderophore production in the focal pathogens (either through phenotypic assays / transcriptomics)? Could such a phenotypic plastic response result in an increased capacity to scavenge iron from the environment? Importantly, increased expression of siderophores has been shown to enhance pathogen virulence (e.g. Lear et al 2023: increased pyoverdine production is linked with increased virulence in *Pseudomonas aeruginosa*). I really appreciate the amount of work the authors have put into this study, but I would suggest expanding the discussion a bit to include a few sentences on(1) unintentional consequences of pyoverdine treatment (e.g. changes in gene expression and non-siderophore-related mutations (e.g. biofilm formation)) on disease dynamics/pathogen virulence:(2) the efficacy of siderophore treatment under more natural conditions, i.e. when the pathogens have to compete with other species in the resident community (i.e. any other effects than resistance evolution through HGT of pyoverdine receptors as mentioned).

Response 1: We would like to thank reviewer # 1 for the positive and constructive assessment. We agree that discussing the above points is important. We have added new paragraphs in the discussion, in which we elaborate on unintentional consequences (lines 532-551) and HGT of receptors (lines 599-607).

**Reviewer #1 (Recommendations For The Authors):**
I only have minor comments/suggestions for the authors, all listed below:• The authors' findings show that the antibacterial activity of pyoverdine is highly context-dependent. As such, I would suggest somewhat toning down the quite general statement in the Abstract: 'Thus, pyoverdines from environmental strain could become new sustainable antibacterials against human pathogens'

Response 2: We agree that the pyoverdine treatment is especially potent against *Acinetobacter baumannii* and *Staphylococcus aureus*, but less so against *Klebsiella pneumoniae*. The treatment success is pathogen-dependent, and we have thus modified the phrase in the abstract (lines 32-34). The new sentence now reads: 'Thus, pyoverdines from environmental strains have the potential to become a new class of sustainable antibacterials against specific human pathogens.' Also in other parts of the manuscript (Results and Discussion), we emphasize that the pyoverdine treatment will likely be effective against specific pathogens (e.g., those with lower-iron affinity siderophores).

• Bacteria often produce more than one type of siderophore. Do you know whether the 320 natural isolates used in this study produce any non-pyoverdine siderophores? Previous work has shown that pyochelin production is suppressed in PAO1 under a wider range of lab conditions. Do you know whether this is the case for the natural isolates used here (and rule out a potential role of non-pyoverdines in iron starvation as observed in Figure 1).

Response 3: This is a valid question. Our own bioinformatic and phenotypic assays reveal that a certain fraction of strains (~ 40%) can produce secondary siderophores (unpublished data). We now mention the existence of secondary siderophores on lines 97-100 and 123. However, we do not think that their contribution to the supernatant assay results is large since the expression of pyoverdine typically suppresses the expression of the secondary siderophores (Cornelis 2010 *Appl Microbiol Biotechnol*; Dumas et al. 2013 *Proc B*) under stringent iron limitation. Furthermore, secondary siderophores have lower iron-binding affinities than pyoverdine. Finally, both the semi-pure and ultra-pure pyoverdine extracts showed strong pathogen inhibition (Fig. 3), and we are thus confident that pyoverdine is responsible for the observed growth inhibition.

• Upon first mentioning the 'mock control' in the Results section in the main text, please state what the actual treatment is.

Response 4: Thank you for noticing this. We now explain in more detail the actual treatment conditions used on lines 103-107 and in the caption of Figure 1. We have further removed the term 'mock' as it is confusing in this context and simple refer to the 'control treatment' in the text.

• Please mention what the different colours mean in the legend of growth recovery in Figure 1B

Response 5: We have clarified the colour scheme in the legend of Figure 1B.

• Please clarify whether you used 12 or 14 strains of human pathogens (the latter number is mentioned in the results section)?

Response 6: In the methods (lines 647-650), we now clearly specify that we used 12 strains of human pathogens in the initial supernatant screen (Figure 1). For all subsequent analyses (dose-response curves and infection experiments), we included the ESKAPE pathogens *K. pneumoniae* and *A. baumannii*.

• Please explain whether ferribactin can be used in any other way than iron chelation (e.g. can this precursor be recycled to form pyoverdine)?

Response 7: We apologize for not having properly explained the role of ferribactin. Under natural conditions, ferribactin is not secreted. It is kept in the periplasmic space, where it matures to pyoverdine. We most likely recovered ferribactin in the supernatant because of the vigorous shaking and centrifugation involved in the pyoverdine purification protocol. We now explain this on lines 216-218. Thus, there is no ferribactin secretion and recycling.

• Have the authors looked at whether there is a relationship between the degree of growth arrest and phylogenetic distance? Would you expect there to be one?

Response 8: This is an interesting question. We have now constructed a phylogenetic tree to explore this relationship (new Figure S2). We found that strains with inhibitory supernatants were scattered across the phylogenetic tree (described on lines 129-135). However, we also found two branches on the tree on which strains with inhibitory supernatant effects were overrepresented. This matches well our previous analysis that closely related species can produce similar pyoverdine types, but that the same pyoverdine can also be produced by completely different species (Gu et al. 2024 *eLife*).

• In the Methods section, please mention you used pyoverdine-only controls in the infection assay.

Response 9: We now mention the use of pyoverdine-only controls in the Methods section (lines 788-790). Overall, we have improved the infection procedure section (starting on line 770). Thank you for pointing this out.

• Did you confirm whether the addition of pyoverdine resulted in lower bacterial loads in Galleria? In other words, were the observed changes in mortality solely related to changes in bacterial density?

Response 10: Thank you for this valid question. No, we did not test whether pyoverdine treatment reduces the bacterial load. However, we did this in the past in two studies with a similar set of pathogens (Weigert et al. 2017 *Evol Appl*; Schmitz et al. 2023 *Proc B*) and found strong correlations between *G. mellonella* survival and bacterial loads. We agree that it is important to understand how pyoverdine affects pathogen load in the host and we will address this point in future studies.

• In your infection assay, were Galleria (n = 10) for each treatment housed in the same environment/container? If so, can you treat these as independent observations or should you use some sort of grouping variable in your survival analysis?

Response 11: Thank you for pointing this out. We forgot to clarify this in the Methods section and now do so on lines 777-779. All larvae were individually housed in separate wells of a 24-well plate. There was no physical contact between larvae and no opportunity for pathogen exchange. As such, we treat each individual larvae as an independent observation.

**Reviewer #2 (Public Review):**
In this work, Vollenweider et al. examine the effectiveness of using natural products, specifically molecules that chelate iron, to treat infectious agents. Through the purification of 320 environmental isolates, 25 potential candidates were identified from natural products based on inhibition assays and were further screened. The structural information and chemical composition were determined.The paper is well-structured and thorough; targeting virulence factors in this manner is a great idea. My enthusiasm is dampened by the mediocre effects of the compounds. The lack of a dose-response curve in the survivability assays suggests a limited scope for these molecules. While it is encouraging that the best survivability occurred at the lowest toxicity level, it opens questions as to how effective such molecules can be. Either the reduction in mortality was offset by using higher concentrations, which was not observed in the compound-alone test, or there is no dose-response curve. The latter would suggest to me that the variation in survivability is not due to the addition of siderophores.

Response 12: Thank you very much for the overall positive assessment. We understand your concern regarding the effectiveness of pyoverdines in the host. However, we wish to emphasize that hazard risks were reduced by more than 50% when treating *A. baumannii* and *K. pneumoniae*. Moreover, it was not so surprising to us that the treatment worked best at intermediate pyoverdine concentrations. We anticipated that pyoverdines could have negative effects for the host at relatively high concentrations because siderophore can interfere with host iron stocks (see discussion starting on line 552). Finally, dose-response curves do not necessarily need to be linear or sigmoid, they can also be hump-shaped. To better illustrate this aspect, we have now plotted the time to death for all the deceased larvae against the pyoverdine concentration gradient and fitted polynomial regression (new Fig. S6). For the above two pathogens, we found humped-shaped dose-response curves in four out of the six comparisons. We present this new analysis on lines 351-362.

I would also like to see how these molecules compare to other iron-chelating molecules. Desferoxamine is a bacteria-derived siderophore that is FDA-approved. However, it is not used to treat infections. Would the author consider comparing their candidate molecules to well-studied molecules? This also raises questions about the novelty of this work; I think the authors could rephrase the discussion to better reflect that bioprospecting for iron-chelating molecules has previously occurred and been successful.

Response 13: Thank you for the comment. The initial version of our manuscript already featured a brief discussion on other iron-chelation therapies. We have now changed the narrative to better reflect the differences of our approach to already existing iron-chelating molecules such as deferoxamine (lines 608-632).

Finally, I am concerned about the few mutations reported in the resistance study. Looking at the SI, it appears that very few mutations were seen. It is unclear what filtering the authors used to arrive at such a low number of mutations. Even filtering against mutations that were selected by adaptation to the media, it seems low that only a handful of clones had distinct mutations.

Response 14: We apologise for the unclear explanations and data analysis. When reanalysing the data we indeed detected a mistake: we originally treated all genomes as clonal origin, despite the fact that we sequenced entire populations for the control treatments. We have now completely re-done the mutational analysis using the breseq pipeline as newly described in the Methods (lines 861-866) and presented in the Results (lines 421-451). We have improved the filtering process and indeed found many more mutations, including the loss of mobile genetic elements. However, it is important to note that it is not uncommon to only find a few beneficial mutations. Especially, in cases where there are selective sweeps often only a few mutations fix.

This paper has a lot of strengths. The workflow is logical and well-executed; the only significant weakness is the effect of the molecules and the lack of an explanation for a dose-response curve in the survivability assay, especially when compared to the data reported in Figure 3. As the authors describe in lines 214-217.

Response 15: Thank you for this overall positive assessment. As discussed in our response 12, the effect of the molecule in the host was not weak as it decreased hazard risks by more than 50% for *A. baumannii* and *K. pneumoniae*. Moreover, we explain that the benefit of the pyoverdine treatment (in terms of treating the infection) can be offset by adverse effects on the host, especially at high pyoverdine concentrations.

**Reviewer #2 (Recommendations For The Authors):**
• Compare these compounds to well-studied iron chelating molecules.

Response 16: We have addressed this comment in our response 13.

• Considering adding time of death to the analysis for the survivability. While the reduction in mortality was not large perhaps the time to death increased.

Response 17: This is an excellent suggestion. We have now analysed the time-to-death as a function of pyoverdine concentration (new Figure S6). Time-to-death was highly variable and sample size was fairly low for *A. baumannii* and *K. pneumoniae* as many larvae survived. Nonetheless, we found hump-shaped dose-response curves in four out of six comparisons and a linear dose-response curve in one case. We now report the new analyses on lines 351-362. Finally, we like to stress once more that reduction in mortality was considerable (hazard risk reduction by more than 50%).

• I would also like to see the actual growth curves of the pathogens in the SI to accompany Fig 6.

Response 18: This is a good point. We have now included the actual growth curves of the pathogens in the Supporting Information to accompany Figure 6 (new Figures S9 and S10).